# Structure-guided conversion from an anaplastic lymphoma kinase inhibitor into *Plasmodium* lysyl-tRNA synthetase selective inhibitors
Jintong Zhou[1,2,5], Mingyu Xia[2,5], Zhenghui Huang[3], Hang Qiao[2], Guang Yang[1], Yunan Qian[3], Peifeng Li[2], Zhaolun Zhang [4], Xinai Gao[4], Lubin Jiang[3], Jing Wang [1,2 ✉], Wei Li [4 ✉] & Pengfei Fang [1,2 ✉]

Aminoacyl-tRNA synthetases (aaRSs) play a central role in the translation of genetic code, serving as attractive drug targets. Within this family, the lysyl-tRNA synthetase (LysRS) constitutes a promising antimalarial target. ASP3026, an anaplastic lymphoma kinase (ALK) inhibitor was recently identified as a novel *Plasmodium falciparum* LysRS (*Pf*LysRS) inhibitor. Here, based on cocrystal structures and biochemical experiments, we developed a series of ASP3026 analogues to improve the selectivity and potency of LysRS inhibition. The leading compound 36 showed a dissociation constant of 15.9 nM with *Pf*LysRS. The inhibitory efficacy on *Pf*LysRS and parasites has been enhanced. Covalent attachment of L-lysine to compound 36 resulted in compound 36K3, which exhibited further increased inhibitory activity against *Pf*LysRS but significantly decreased activity against ALK. However, its inhibitory activity against parasites did not improve, suggesting potential future optimization directions. This study presents a new example of derivatization of kinase inhibitors repurposed to inhibit aaRS.

Antimicrobial resistance, encompassing both bacteria and malaria, has become a major public health threat worldwide. In recent decades, the number of new antibiotics entering the market has decreased significantly due to scientific, regulatory, and economic barriers[1]. This necessitates the development of novel antimicrobial compounds with innovative chemical scaffolds and mechanisms of action to conquer resistant pathogens.

Aminoacyl-tRNA synthetases (aaRSs) catalyze the synthesis of aminoacyl-tRNAs, which are crucial building blocks for protein biosynthesis. These enzymes are essential for all cellular life, playing a central role in the translation of the genetic code[2,3]. As the indispensable components of the protein translation apparatus, aaRSs have been considered as attractive targets for the development of antiparasitic, antifungal, antibacterial agents and for the treatment of other human diseases[4–10].

The aminoacylation reactions catalyzed by different aaRSs follow a similar two-step process, involving the binding and activation of the cognate amino acid with ATP, followed by transfer of the activated amino acid to the 3′-end of tRNA. Correspondingly, the active center of aaRSs has three adjacent subsites that bind to ATP, amino acid, and tRNA, respectively[11]. These subsites are considered as hot spots in the development of aaRS inhibitors. Mupirocin, an isoleucyl-tRNA synthetase (IleRS) inhibitor, occupies the isoleucine and ATP binding sites in the IleRS active center to inhibit Gram-positive aerobic bacteria and a few Gram-negative strains[4,12]. It is bacteriostatic against *Staphylococcus aureus* at concentrations ranging from 0.05 to 0.5 μg·mL$^{-1}$ (MIC is 0.05 μg·mL$^{-1}$) and it is bactericidal at higher concentrations[12–14]. The febrifugine derivative Halofuginone (HF), a prolyl-tRNA synthetase (ProRS) inhibitor, binds to the proline and tRNA A$^{76}$ sites[8]. It has been used as a veterinary drug against coccidiosis and received an orphan drug designation for the treatment of human scleroderma from the Food and Drug Administration (FDA) in 2000[15]. HF inhibits *P. falciparum* proliferation in cellulo, selectively binds and inhibits

[1]School of Chemistry and Materials Science, Hangzhou Institute for Advanced Study, University of Chinese Academy of Sciences, 1 Sub-lane Xiangshan, Hangzhou 310024, China. [2]State Key Laboratory of Chemical Biology, Shanghai Institute of Organic Chemistry, University of Chinese Academy of Sciences, Chinese Academy of Sciences, 345 Lingling Road, Shanghai 200032, China. [3]Key Laboratory of Molecular Virology and Immunology, Shanghai Institute of Immunity and Infection, Chinese Academy of Sciences, 320 Yueyang Road, Shanghai 200031, China. [4]Department of Medicinal Chemistry, School of Pharmacy, China Pharmaceutical University, 639 Longmian Avenue, Nanjing, Jiangsu 211198, China. [5]These authors contributed equally: Jintong Zhou, Mingyu Xia. ✉e-mail: jwang@sioc.ac.cn; wli@cpu.edu.cn; fangpengfei@sioc.ac.cn

recombinant $Pf$ProRS within the low nanomolar range[16,17]. Additionally, HF has potential therapeutic applications in cancer and fibrotic diseases which are being tested in clinical trials[18]. It can inhibit the proliferation and metastasis of tumor cells in several cancer models[19–21]. It was also reported to limit SARS-CoV-2 entry by reducing surface expression of TMPRSS2[22]. In short, the development of aaRS inhibitors have been widely explored in recent years as potential therapeutic agents.

Lysyl-tRNA synthetase (LysRS) has been validated as a promising target for antimalarial, anti-*Cryptosporidium*, anti-*Pseudomonas aeruginosa*, and anti-*Mycobacterium tuberculosis* therapies[23–25]. The ATP-binding pocket is the most promising area for the development of LysRS inhibitors. Cladosporin, a potent LysRS inhibitor, effectively inhibits *P. falciparum* LysRS ($Pf$LysRS) activity by mimicking the interaction of ATP[26]. Further research is underway to develop more ATP-competitive inhibitors of $Pf$LysRS to create new chemical scaffolds with improved druggability[25,27–31].

ASP3026, an anaplastic lymphoma kinase (ALK) inhibitor, was initially used in clinical trials for the treatment of B-cell lymphoma and solid tumors[32]. It has ideal metabolic stability and excellent oral absorption in the recently completed phase I clinical trial[32]. Recently, we found that ASP3026 is also a potent $Pf$LysRS inhibitor with an $IC_{50}$ in the submicromolar range[33]. ASP3026 occupies the ATP binding site of ALK or $Pf$LysRS to prevent their binding with ATP. Normally, ALK is an important kinase in the development of the central nervous system[34–36]. ASP3026 exhibits strong inhibition activity against ALK with an $IC_{50}$ of 3.5 nM[33,37]. To develop anti-infective molecules targeting LysRS, we aim to decrease their inhibitory activity against ALK to avoid potential adverse side effects.

Here, we develop a series of ASP3026 analogues that switch the targeting specificity from ALK to LysRS. Based on the analysis of cocrystal structures, we reconstructed the compounds to fit the ATP binding site of LysRS instead of ALK. These compounds were evaluated by thermal shift, enzyme activity assays and *Plasmodium* growth inhibition experiment. We determined five cocrystal structures of LysRS in complex with the compounds in order to provide a basis for further optimization. We observed that the substitution of 4-(4-methylpiperazinyl)piperidinyl moiety can decrease the inhibition against ALK. The 5-methoxy-substituted compound **36** showed stronger inhibition against $Pf$LysRS in the ATP-hydrolysis assay and the erythrocytic-stage *Plasmodium* growth inhibition experiment. We linked L-lysine to compound **36** and obtained compounds **36K2–36K5**, which further enhances the LysRS inhibitory activity while removing the ALK inhibitory activity. Overall, our studys provide insights into converting kinase ATP site inhibitors to aaRS inhibitors.

## Results
### Analysis of action modes of ASP3026 on PfLysRS and ALK
Binding of small molecules to the ATP site of $Pf$LysRS can significantly enhance the protein's thermal stability[25,30,31,38]. Based on this property, we designed a high-throughput screening experiment and found that compound ASP3026 significantly improves the thermal stability of $Pf$LysRS. Furthermore, it was verified to be an ATP competitive inhibitor of $Pf$LysRS[33]. ASP3026 was initially developed to inhibit ALK by competing with ATP. It has a strong inhibitory activity against ALK with an $IC_{50}$ of 3.5 nM[37], while the $IC_{50}$ for inhibiting $Pf$LysRS enzyme activity is in the submicromolar range (Supplementary Fig. 1)[33]. Efforts were made to modify its selectivity for LysRS inhibition.

NVP-TAE684, a type 1 inhibitor of ALK, is another ATP competitive inhibitor with an inhibition constant (Ki) of 0.65 nM[39], and it is very similar in structure to ASP3026 (Fig. 1a). Because the ALK/ASP3026 complex structure has not been reported, we compared the ALK/NVP-TAE684 complex structure (PDB: 2XB7) with the $Pf$LysRS/ASP3026 complex structure (PDB: 7BT5) to identify potential compound optimization strategies[33,40].

First, the triazine (corresponds to pyrimidine in NVP-TAE684) appears to be essential for ASP3026 to interact with both proteins. It occupies the adenine site in the complex structure of both proteins (Fig. 1b–d). In $Pf$LysRS, the triazine of ASP3026 replaces adenine of ATP to form stacking interaction with Phe342, and two nitrogen atoms of the azine and the imine forms two H-bonds with the main chain atoms of Asn339 (Fig. 1e). In ALK, the pyrimidine of NVP-TAE684 formed hydrophobic interactions with Leu1122, Ala1148, Leu1198, Met1199, and Leu1256, and two nitrogen atoms of the pyrimidine and the imine forms two H-bonds with the main chain atoms of Met1199 (Fig. 1f). Second, the isopropyl sulfonate moiety binds to the two proteins by mimicking different parts of ATP. In $Pf$LysRS, it mimics the ribose of ATP to form van der Waals interactions with Val500, Glu501, and Gly556 (Fig. 1g). In ALK, it is located near the position of ADP $\beta$-phosphate and forms two H-bonds with Lys1150 (Fig. 1h). The 4-(4-methylpiperazine) piperidine group binds to the enzyme in two completely different conformations, using ATP as a reference (Fig. 1d). In $Pf$LysRS, it protrudes from the pocket of $Pf$LysRS and forms van der Waals interactions with Glu332, Asp335, and His338 (Fig. 1i). In ALK, this part of the compound also sticks out of the pocket of ALK, but in a more extended conformation, and forms van der Waals interactions with Asp1203, Ser1206, and Glu1210 (Fig. 1j).

With these analyses, we first optimized the 4-(4-methylpiperazine) piperidine group, which has the most distinct effect on the two proteins, then modified and optimized the triazine and sulfonyl aniline moieties.

### Substitution of the 4-(4-methylpiperazinyl)piperidinyl moiety significantly improved the selectivity against *Pf*LysRS in vitro
Compound **33–42** were synthesized by replacing of the original 2-methoxy-4-(4-methylpiperazinyl)piperidinyl aniline with the commercially available corresponding anilines (Supplementary Fig. 2, Supplementary Table 2). Compared with ASP3026, compound **38** has removed the 4-(4-methylpiperazine) piperidine, which is probably less useful for binding LysRS, and left the rest unchanged (Fig. 2a). In the thermal shift experiments, **38** increased the Tm value of $Pf$LysRS by approximately 9.0 °C in the presence of L-lysine (Fig. 2b). This similar result to ASP3026 (Supplementary Table 2) indicates that the binding ability of the compound to $Pf$LysRS remained unchanged after the removal of this group. Consistently, the $IC_{50}$ of **38** for inhibiting $Pf$LysRS activity was 910 nM (Fig. 2c), slightly higher than ASP3026 whose $IC_{50}$ was 965 nM in this assay (Supplementary Fig. 1). We also performed surface plasmon resonance (SPR) experiments and sensorgram analysis for the interactions of **38** with immobilized $Pf$LysRS resulted in a dissociation constant ($K_D$) of 62.8 nM (Supplementary Fig. 3a). It confirmed that the 4-(4-methylpiperazinyl)piperidinyl moiety is not essential for $Pf$LysRS binding. However, the activity of compound **38** against ALK with the $IC_{50}$ value of 532 nM (Fig. 2d) was significantly reduced compared with ASP3026[33,37]. Compound **38** maintains a high species selectivity in vitro. Its selectivity of $Pf$LysRS over human LysRS ($Hs$LysRS) reached 328 times (Fig. 2c, e). Therefore, consistent with crystal structure analysis, this modification reduces the ability of the compound to inhibit ALK while maintaining its $Pf$LysRS inhibitory activity.

In comparison to compound **38**, compound **36** has one more methoxy group substitution in the para position of the original 2-methoxy group (Fig. 3a). Compound **36** exhibited the highest $Pf$LysRS binding affinity among these ASP3026 analogues (Supplementary Table 2). The ΔTm of $Pf$LysRS induced by compound **36** was approximately 11.8 °C in the presence of L-lysine. It also improved the Tm of $Pf$LysRS by 11.7 °C in the absence of L-lysine (Fig. 3b), indicating that the binding of compound **36** to $Pf$LysRS is independent of the presence of L-lysine. This property is similar to ASP3026, but different from cladosporin[41]. The $K_D$ value of compound **36** was 15.9 nM in SPR experiments (Supplementary Fig. 3b), indicating stronger binding affinity than compound **38**. Compound **36** also showed strong inhibition of $Pf$LysRS enzyme activity in the ATP hydrolysis assay, with an $IC_{50}$ of 198 nM (Fig. 3c), which was also stronger than ASP3026 in vitro. The ALK inhibitory ability of **36** was significantly reduced, and the $IC_{50}$ was greater than 19.1 μM (Fig. 3d). This result indicated that the increase of methoxide at this location may bring repulsion with the carboxyl group of Asp1203 (Fig. 1j), which is not favorable for the interaction of the compound and ALK. At the same time, the inhibition of the compound on $Hs$LysRS was also significantly weakened (Fig. 3e).

**Fig. 1 | Comparison of crystal structures between *Pf*LysRS/ASP3026 and ALK/NVP-TAE684.**
**a** Chemical structures of ASP3026 and NVP-TAE684. **b** Zoom-in view of ASP3026 localization in the conserved ATP binding site of *Pf*LysRS (PDB: 7BT5). ASP3026 and ATP are depicted as sticks. **c** Zoom-in view of NVP-TAE684 localization in the conserved ATP binding site of ALK (PDB: 2XB7). ASP3026 and ADP are depicted as sticks. **d** ASP3026 in *Pf*LysRS and NVP-TAE684 in ALK are superimposed using the adenine ring of ATP as a reference. **e** The triazine of ASP3026 replaces adenine of ATP to form stacking interaction with Phe342, and forms two H-bonds with the main chain atoms of Asn339 in *Pf*LysRS. **f** The chloropyrimidine of NVP-TAE684 formed hydrophobic interactions with Leu1122, Ala1148, Leu1198, Met1199, and Leu1256, and forms two H-bonds with the main chain atoms of Met1199. **g** The isopropyl sulfonate moiety of ASP3026 mimics the ribose of ATP to form van der Waals interactions with Val500, Glu501, and Gly556 of *Pf*LysRS. **h** The isopropyl sulfonate moiety of NVP-TAE684 is located near the position of ADP β-phosphate and forms two H-bonds with Lys1150 of ALK. **i** In *Pf*LysRS, the 4-(4-methylpiperazine) piperidine group protrudes from the pocket of *Pf*LysRS and forms van der Waals interactions with Glu332, Asp335, and His338. **j** In ALK, the 4-(4-methylpiperazine) piperidine group sticks out of the pocket of ALK in a more extended conformation, and forms van der Waals interactions with Asp1203, Ser1206, and Glu1210.

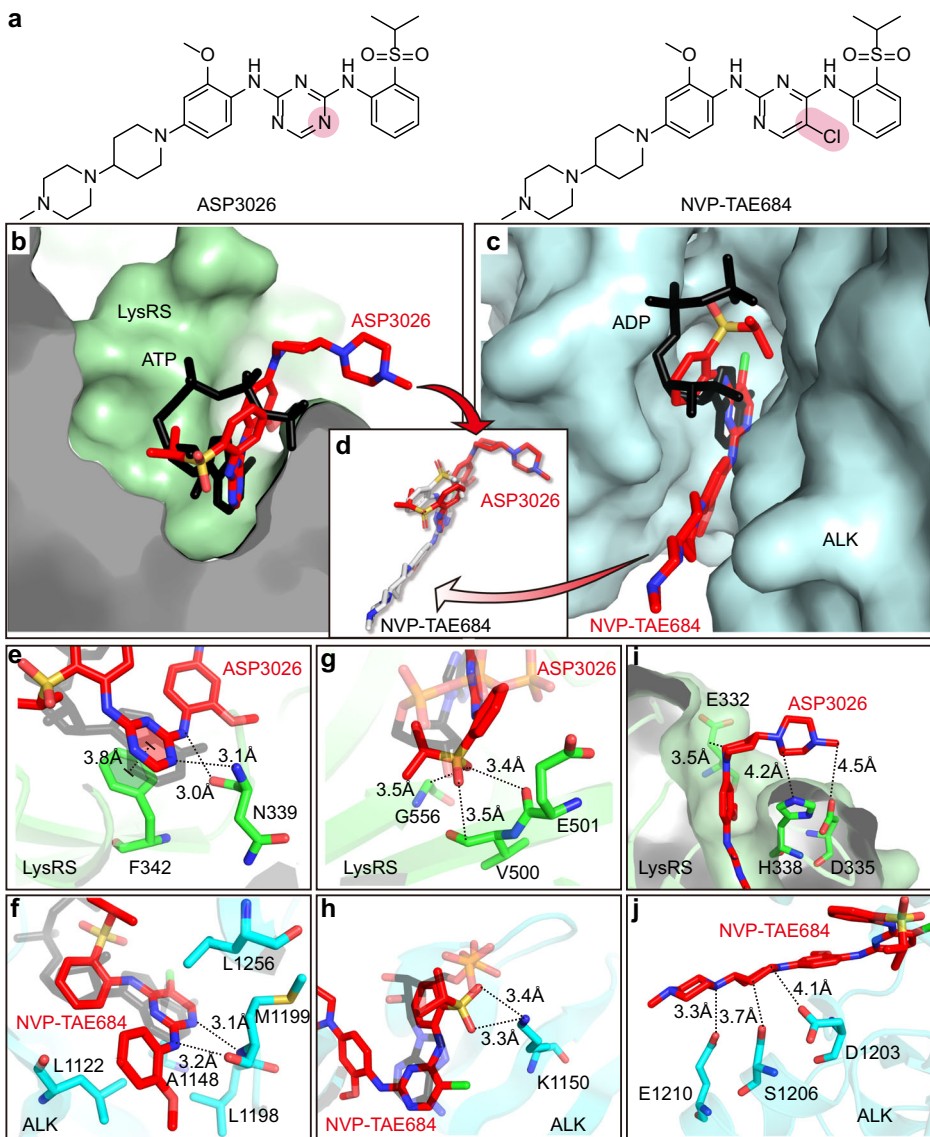

Compared to compound **36**, the methoxy group in the amino meta position of compound **34** is replaced by a methoxycarbonyl group, and the substitution position of methoxy group in compound **35** is different (Supplementary Fig. 4a). As a result, the ΔTm values of *Pf*LysRS caused by compound **34** and **35** were both lower than that caused by compound **36** (Supplementary Fig. 4b, c), and the potency of compound **34** against *Pf*LysRS was weaker than that of compound **36** (Supplementary Fig. 4d).

To understand how these compounds are recognized by *Pf*LysRS, we determined the cocrystal structures of *Pf*LysRS with the compounds **34, 35, 36**, and **38** (Supplementary Table 1). Most of the interactions between the compounds and the protein were preserved compared to *Pf*LysRS-ASP3026 (Figs. 1, 4). The methoxy group inherited from ASP3026 in these compounds were all bound to the hydrophobic cavity formed by Arg330, Glu332, His338, Asn339, and Pro340 like ASP3026. The additional methoxy group of compound **36** formed a new H-bond with Arg330 (Fig. 4a). Similarly, the carbonyl oxygen of compound **34** also formed a new H-bond with Arg330 (Fig. 4b). Without these substituent groups or with a different substitution position, compound **38** and **35** lacked this H-bond (Fig. 4c, d), consistent with their weaker activity compared to compound **36** (Figs. 2, 3, Supplementary Fig. 4, Supplementary Table 2). These compounds occupy a space in the ATP binding pocket of LysRS and overlap mostly with cladosporin (Supplementary Fig. 5). Most of the strongly interacting residues

are conserved between *Pf*LysRS and *Hs*LysRS (Supplementary Fig. 5). It has been reported that three different residues (Thr307, Val328, and Ser344) and the unique skeleton dynamics make *Pf*LysRS more sensitive to cladosporin than *Hs*LysRS[26,42]. The selectivity of these Asp3026 derivatives to *Pf*LysRS may be for the same reason.

### Replacement of aromatic ring did not show significant positive effects

Then we replaced the aromatic rings in the structure of the compound and evaluated their activity and selectivity. First, the methoxyaniline group of compound **38** were replaced with pyridine ring substrates to obtain compounds **38a** and **38b**, differing from compound **38** by only one atom (Fig. 5a). Surprisingly, there was a significant difference in the ΔTm value of *Pf*LysRS between the two compounds, solely due to the different position of the N atom in the pyridine ring. The ΔTm of **38a** was 7.9 °C while the ΔTm of **38b** was only 2.0 °C (Fig. 5b, Supplementary Table 3). It indicates that compound **38a** has a similar binding ability to *Pf*LysRS as **38**, while **38b** can only weakly bind to *Pf*LysRS. In the ATP hydrolysis assay, **38a** showed slightly stronger inhibition of *Pf*LysRS than **38**, with an IC₅₀ of 568 nM, while its inhibition ability to ALK was weakened compared to **38** (Fig. 5c, d). The IC₅₀ of **38a** against *Hs*LysRS was 84 μM, indicating a significant enhancement in its inhibitory activity compared to **38** (Fig. 5e). The only

**Fig. 2 | The removal of the 4-(4-methylpiperazine) piperidine group preserves *Pf*LysRS inhibitory activity and reduces ALK inhibitory activity in vitro. a** Chemical structure of compound **38**. **b** Diagram of the Tms of *Pf*LysRS in the presence of L-lysine and/or compound **38**. Error bars represent standard deviations (SD) of four technical repeats. **c** The potency of compound **38** against *Pf*LysRS is measured using the ATP hydrolysis assay. **d** The potency of compound **38** against ALK is measured using the ATP hydrolysis assay. **e** The potency of compound **38** against *Hs*LysRS is measured using the ATP hydrolysis assay. Error bars in **c**–**e** represent SD of three technical repeats.

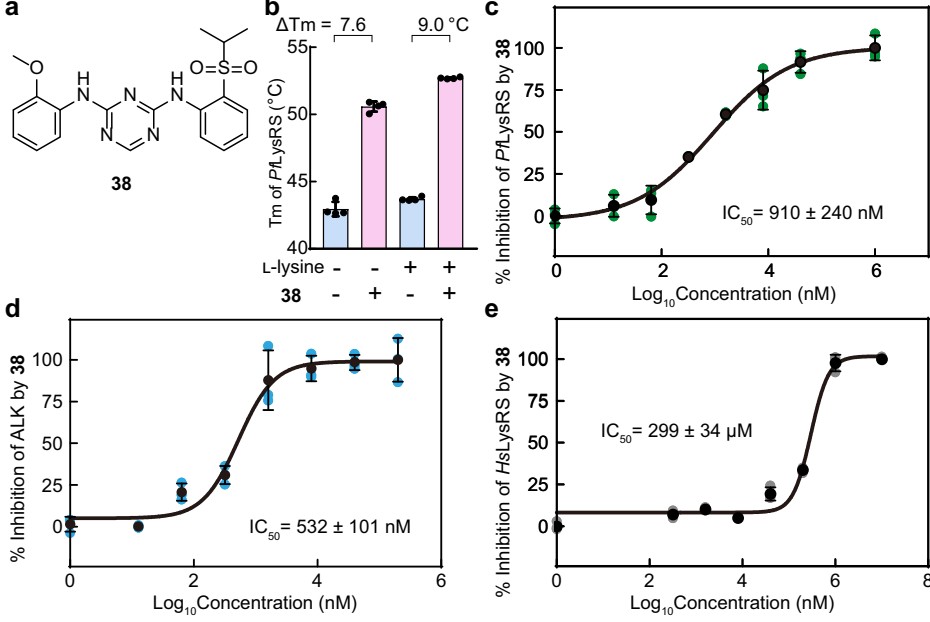

**Fig. 3 | Compound 36 is a potent and highly selective *Pf*LysRS inhibitor. a** Chemical structure of compound **36**. **b** Diagram of the Tms of *Pf*LysRS in the presence of L-lysine and/or compound **36**. Error bars represent SD of four technical repeats. **c** The potency of compound **36** against *Pf*LysRS is measured using the ATP hydrolysis assay. **d** The potency of compound **36** against ALK is measured using the ATP hydrolysis assay. **e** The potency of compound **36** against *Hs*LysRS is measured using the ATP hydrolysis assay. Error bars in **c**–**e** represent SD of three technical repeats.

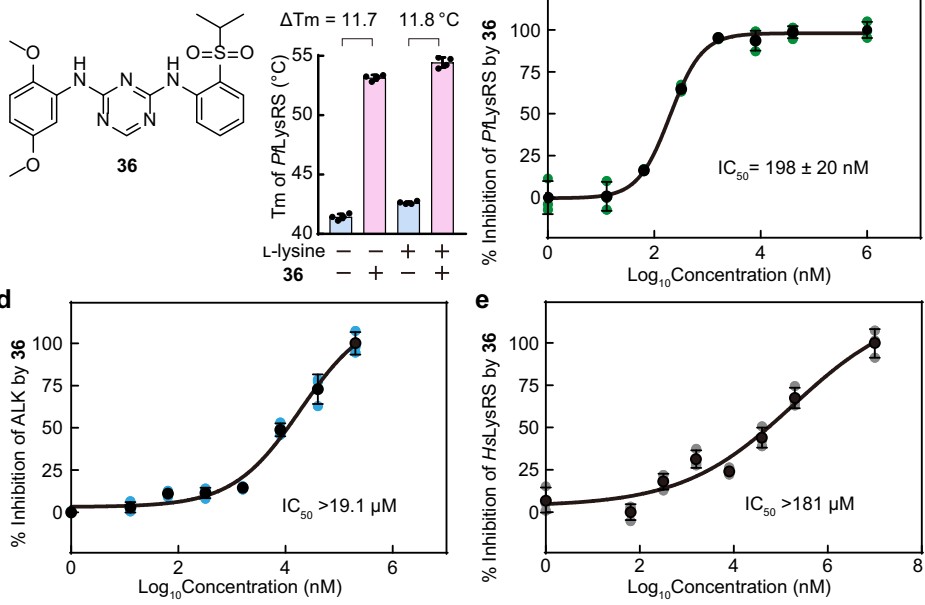

structural difference between these two compounds is that the benzene ring in methoxyaniline block is replaced by pyridine ring. This change enhanced the *Hs*LysRS inhibition activity of **38a** to a certain extent. However, compound **38b**, which also uses a pyridine ring instead of a benzene ring as a substrate, significantly decreased the ability to stabilize *Pf*LysRS in the thermal shift assay. We performed DFT calculations for the local minimum energy conformations of compounds **38a** and **38b** based on the binding conformation of compound **38** in crystal structure to rationalize the activity difference (Supplementary Fig. 6a, b). The local minimum energy conformation of compound **38a** is almost the same as the *Pf*LysRS binding conformation. However, in the local minimum energy conformation of **38b** (Supplementary Fig. 6b), since the C-H becomes N and reduces the repulsion with C-H in the other benzene ring, the three aromatic rings converge to the same plane due to conjugation. There is a significant difference between its minimum energy conformation and the protein binding

conformation ($\Delta G^{pro}$-$\Delta G^{min}$ = 6.1 kcal/mol). Therefore, the minimum energy conformation of **38b** is not conducive to binding *Pf*LysRS.

To assess the impact of replacing the methoxy-linked benzene ring with pyridine, this modification of **38a** was applied to **36**, previously identified as having the highest *Pf*LysRS inhibitory activity. Compound **36a** was then synthesized (Supplementary Fig. 7a, Supplementary Fig. 2). Compound **36a** increased the Tm value of *Pf*LysRS by 10.7 °C in the presence of L-lysine (Supplementary Table 3, Supplementary Fig. 7b), which was comparable to that of **36**. Compared to compound **36**, the *Pf*LysRS inhibitory activity of **36a** was slightly decreased, but the *Hs*LysRS inhibitory activity was significantly enhanced, with an IC₅₀ of 44 μM (Supplementary Fig. 7c, e). Therefore, these results suggest that the replacement of the benzene ring with pyridine has little effect on the inhibitory activity of *Pf*LysRS or ALK (Fig. 5d, Supplementary Fig. 7d), but significantly decreases the species selectivity of the compounds (Fig. 5e, Supplementary Fig. 7e).

**Fig. 4 | Cocrystal structures of *Pf*LysRS with the compounds 34, 35, 36 and 38. a** One methoxy group of compound **36** binds to the hydrophobic cavity formed by Arg330, Glu332, His338, Asn339, and Pro340, while the other methoxy group form an H-bond with Arg330. **b** Compound **34** also formed a new H-bond with Arg330. **c** Without the substituent groups, compound **38** lacked the H-bond with Arg330. **d** With a different substitution position, compound **35** lacked the H-bond with Arg330. Compound **34, 35, 36, 38,** Arg330, Glu332, His338, Asn339, Pro340 and ASP3026 are depicted as sticks.

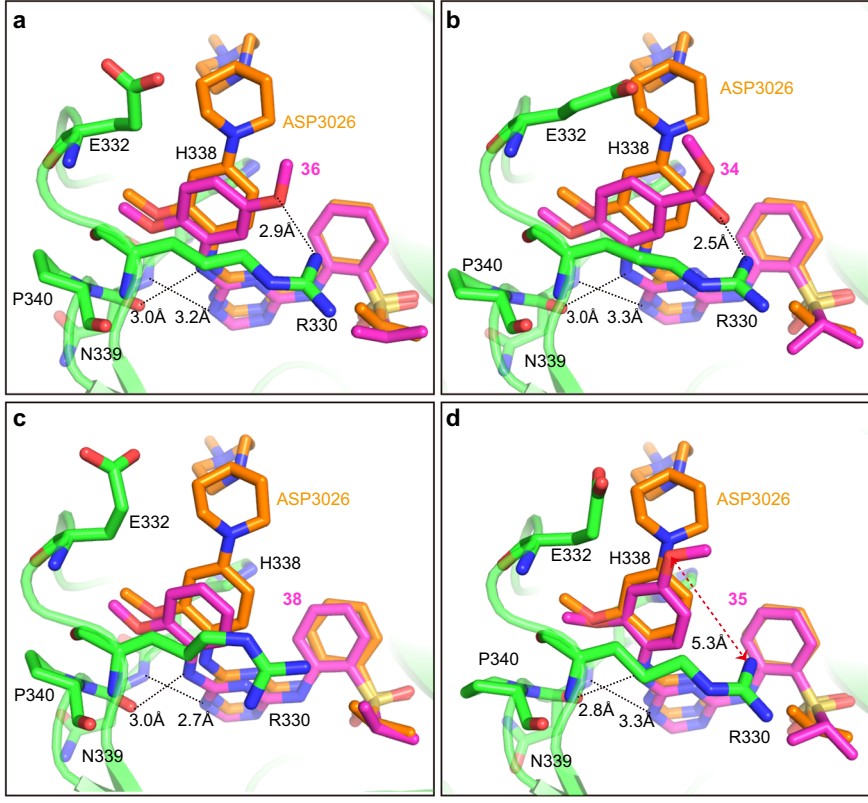

Subsequently, attempts were made to replace the triazine in ASP3026 structure with other aromatic rings. We previously found that NVP-TAE684, whose triazine part is replaced, showed a significant reduction in LysRS binding[33]. In this study, compounds **36b–f** were synthesized using Buchwald–Hartwig reaction (Supplementary Table 4). Surprisingly, these modifications almost completely abolished the binding affinity of the compounds to *Pf*LysRS (Supplementary Table 4). Based on the crystal structure, these changes may weaken or even lose hydrogen bonding interactions with Asn339 residues and stacking interactions with Phe342 residues. Some changes such as changes in **36e** will also increase the repulsion between aromatic rings (Supplementary Fig. 6c), affect the conformation between the three aromatic rings, resulting in the reduced binding. Thus, the chemical synthesis and biochemical results confirmed that the triazine part is essential for this series of compounds to bind LysRS.

## Substitution of isopropylsulfonyl moiety to create synergistic binding

In the thermal shift assays, we did not observe a significant synergistic effect between L-lysine and ASP3026 or ASP3026 analogues (Figs. 2b, 3b and 5b)[33]. The previously reported synergistic effect between cladosporin and L-lysine might depend not only on the co-occupation of ATP and L-lysine pockets (Supplementary Fig. 8a), but also on the van der Waals interaction between the inhibitor and L-lysine[41,43]. Therefore, a potential strategy to enhance the effectiveness of ASP3026 against *Pf*LysRS is to replace the isopropyl sulfonyl part with a larger moiety to mimic the van der Waals interaction between cladosporin and L-lysine, enabling the new compound to bind *Pf*LysRS synergistically with L-lysine (Supplementary Fig. 8b). We synthesized compound **32** by replacing the isopropylsulfonyl group with a cyclohexyl group. The o-cyclohexylaniline substrate was prepared with a Cope rearrangement and hydrogenation reaction after a simple amination. Similar to the synthesis of ASP3026[37], the o-cyclohexylaniline reacted with 2,4-dichloro-1,3,5-triazine using N,N-diisopropylethylamine in tetrahydrofuran. The aniline derivative was then introduced using methanesulfonic acid in ethanol to yield compound **32** (Supplementary Fig. 8c,

Supplementary Fig. 9). Meanwhile, taking the intermediate compound **4** as a substrate, we obtained another analogue compound **31** (Supplementary Fig. 8c, Supplementary Fig. 9).

It turned out that the substitution of an isopropylsulfonyl group for a cyclohexyl group in compound **32** or a cyclohex-2-en-1-yl group in compound **31** was not able to improve the binding of the compounds to *Pf*LysRS. Compounds **31** and **32** increased the Tm of *Pf*LysRS by only 2.0 °C and 4.1 °C, respectively, in the presence of L-lysine (Supplementary Fig. 8d). These results suggest that the cyclohexyl or cyclohexene segments in compounds **31** and **32** may be too large. Therefore, we considered to covalently link L-lysine to ASP3026 analogues instead of creating synergic binding between L-lysine and the inhibitors.

## Covalently linking L-lysine to the inhibitor can enhance its inhibition of *Pf*LysRS enzymatic activity

One significant distinction between the binding pockets of LysRS and ALK is that the LysRS's active pocket includes an L-lysine binding site adjacent to the ATP binding site. Therefore, by covalently linking L-lysine to ASP3026 analogues, it is possible to obtain compounds with enhanced LysRS inhibitory activity and reduced ALK inhibitory activity. Considering that **36** has a relatively high *Pf*LysRS inhibitory activity, L-lysine is connected to the sulfonyl group of **36** through varying lengths of carbon chains. The compounds **36K2–36K5** with a connection chain length of 2 to 5 carbons or a four-membered ring was synthesized using Buchwald–Hartwig reaction, followed by hydrogenation to remove the Cbz protection (Fig. 6a, Supplementary Fig. 10).

As expected, the ability of the compounds to bind *Pf*LysRS was significantly enhanced with covalently linked L-lysine. In the thermal shift assay, the ΔTm values of four new compounds were higher than that of **36**, especially **36K3** and **36K4** with an alkyl chain of 3 or 4 carbons (Fig. 6b). Their ΔTm values were 12.8 °C and 12.9 °C, respectively, comparable to cladosporin[41]. The K$_D$ value of **36K3** reached 12.4 nM (Supplementary Fig. 11). Meanwhile, the selectivity of these compounds for LysRS and ALK inhibition was further enhanced by L-lysine connection (Supplementary

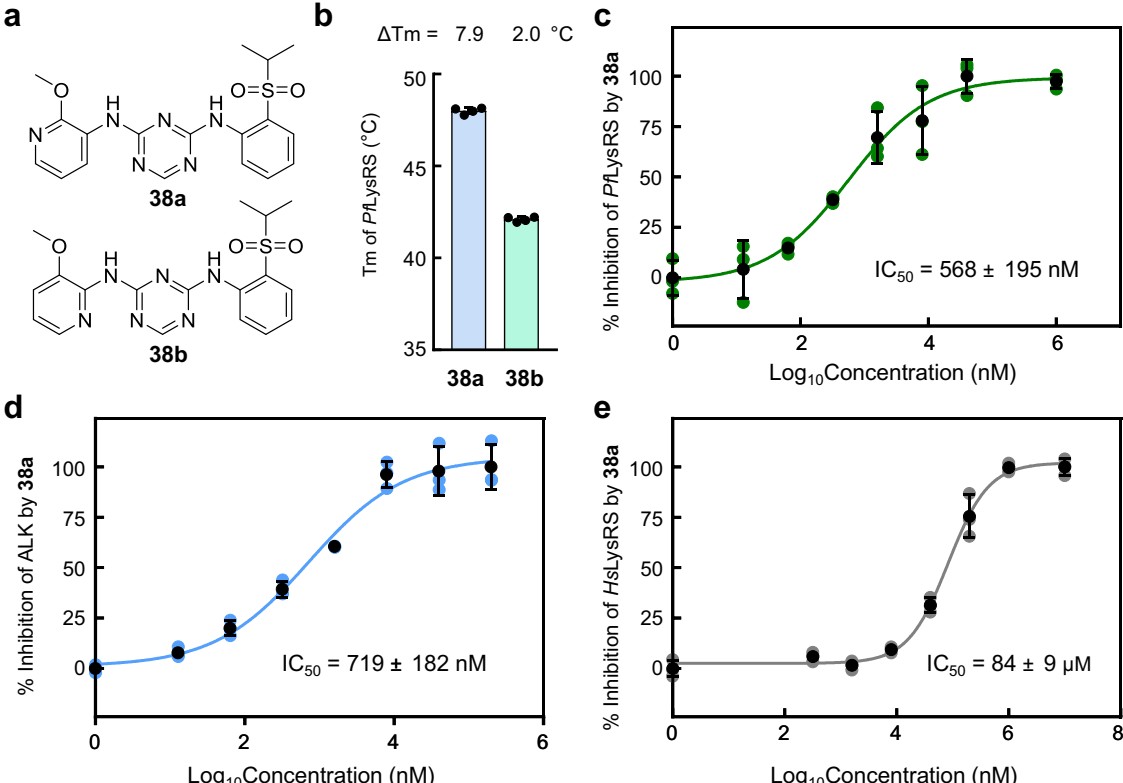

**Fig. 5 | The position of the N atom in the pyridine ring replaced compounds 38a and 38b has a great influence on the activity. a** Chemical structures of compound **38a** and **38b**. **b** Diagram of the Tms of *Pf*LysRS in the presence of L-lysine and compound **38a** or **38b**. Error bars represent SD of four technical repeats. **c** The potency of compound **38a** against *Pf*LysRS is measured using the ATP hydrolysis assay. **d** The potency of compound **38a** against ALK is measured using the ATP hydrolysis assay. **e** The potency of compound **38a** against *Hs*LysRS is measured using the ATP hydrolysis assay. Error bars in **c–e** represent SD of three technical repeats.

Table 5). Compounds **36K2**–**36K4** and **36K4'** showed strong inhibition on the activity of *Pf*LysRS. In particular, the $IC_{50}$ of **36K3** against *Pf*LysRS inhibition reached 59.2 nM (Fig. 6c), displaying an apparent selectivity index of 1250 for ALK and 200 for *Hs*LysRS (Supplementary Table 5).

We also determined the mode of action of **36K3** through crystal structure (Supplementary Table 1). Consistent with our design, compound **36K3** occupies both the ATP-binding site and L-lysine binding site of *Pf*LysRS. The conformation of **36K3** at the ATP binding site is similar to that of **36**, and its L-lysine part generally matches the L-lysine binding mode. Due to its covalent connection to compound **36**, the L-lysine moiety is unable to adjust its conformation freely, so it is pulled out of its original binding site a little bit (Fig. 6d). Therefore, further enhancement of the compound's activity is possible through a better linkage between **36** and L-lysine.

### Some ASP3026 analogues improve the inhibition against the blood stage of *P. falciparum*

ASP3026 inhibited the ATP-hydrolysis activity of *Pf*LysRS at the sub-micromolar level and consistently suppressed the growth of erythrocytic-stage *P. falciparum* 3D7 parasites with an $EC_{50}$ of 5.61 µM[33]. To determine the inhibitory effect of ASP3026 analogues on *Plasmodium* growth, we further evaluated the inhibitory potency of these compounds on *P. falciparum* strain 3D7 in the erythrocytic-stage (Fig. 7a, b, and Supplementary Fig. 12). Among these compounds, compounds **34, 36** and **38** demonstrated greater inhibitory activity on parasite growth compared to ASP3026 (Fig. 7c–e). Compound **36**, in particular, exhibited the most potent inhibition with an $EC_{50}$ value of 736 nM, surpassing the potency of ASP3026 by over sevenfold (Fig. 7d). Meanwhile, compounds **36a** and **38a** have weaker inhibitory effects on the growth of *Plasmodium falciparum* compared to counterparts compounds **36** and **38** (Fig. 7f, Supplementary Fig. 12d, e). It indicates that replacing the benzene ring with the pyridine ring is not

conducive to the inhibition of compound growth on *P. falciparum*. Furthermore, compounds **36K3, 36K4** and **36K4'** showed weaker inhibitory activity in this experiment (Supplementary Fig. 12f, g, h), which was much different from the results of enzymatic activity inhibition experiment.

We estimated the cytotoxicity of the compounds **34, 36 38, 38a, 36a** and **36K3** in human hepatocyte carcinoma HepG2 cells (Supplementary Fig. 13). Compound **38** exhibits an $CC_{50}$ of 7.7 µM (Supplementary Fig. 13c), which may due to its inhibition of ALK ($IC_{50}$ = 532 nM). Consistent with the weak inhibition of compound 36 on ALK and *Hs*LysRS, it was weakly toxic and showed no significant toxicity to HepG2 (Supplementary Fig. 13b). The cytotoxicity of **34, 38a, 36a,** and **36K3** was lower than that of compound **38** but higher than that of compound **36**, which may be related to their mild inhibition of ALK or *Hs*LysRS (Supplementary Fig. 13d, e). Together with the cellular antiparasitic activity, **36** is the most promising lead compound.

The conjugation of L-lysine to a compound elevates its molecular weight and enhances its polarity, potentially impeding its transmembrane permeability. The increased competition of endogenous L-lysine may also be the reason for the decreased inhibition of **36K3** on *P. falciparum* growth. Other strategies using the L-lysine binding sites need to be further developed. Together, the modifications of compound ASP3026 have led to the development of new compounds with improved inhibitory effects against *P. falciparum*.

### Discussion

In this study, we designed and synthesized a series of ASP3026 derivatives. Among them, compound **36** demonstrated over sevenfold enhanced activity in inhibiting *Plasmodium* growth compared to ASP3026, suggesting a potential advancement in antiplasmodial efficacy through structural optimization.

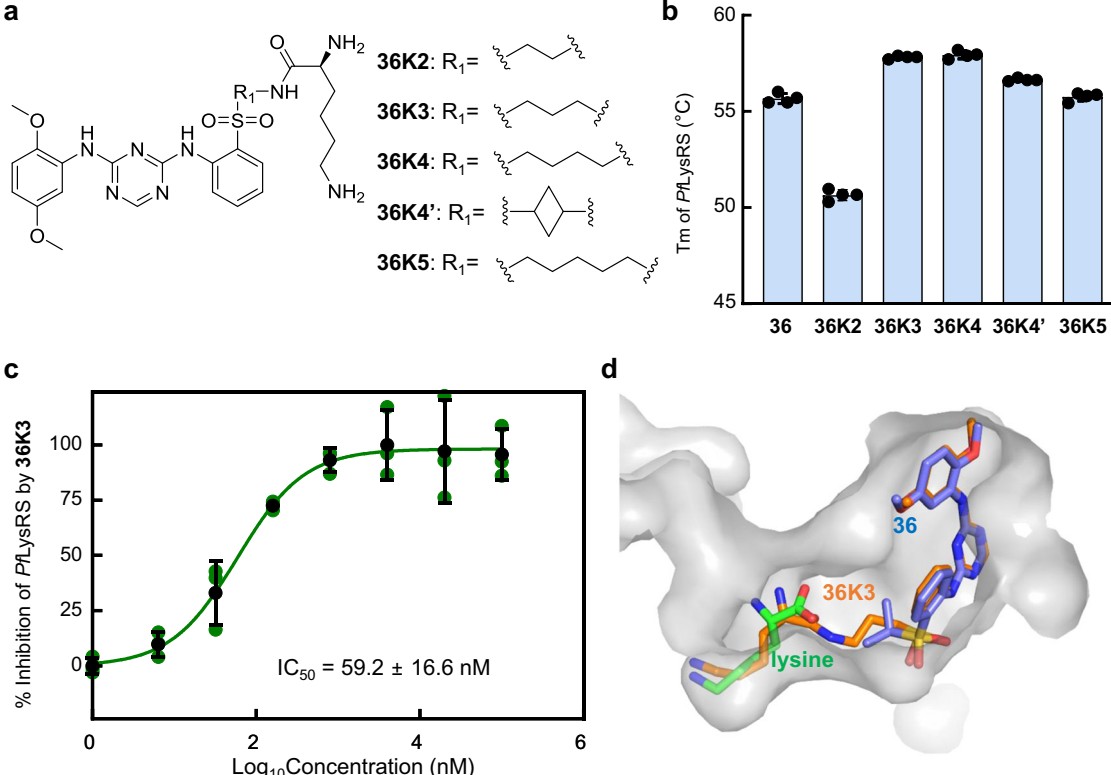

**Fig. 6 | Covalently linking L-lysine to ASP3026 analogue 36 enhances the binding affinity to *Pf*LysRS. a** Chemical structures of compound **36K2–36K5. b** Diagram of the Tms of *Pf*LysRS in the presence of L-lysine and compounds **36K2–36K5.** Error bars represent SD of four technical repeats. **c** The potency of compound **36K3** against *Pf*LysRS is measured using the ATP hydrolysis assay. Error bars represent SD of three technical repeats. **d** Cocrystal structures of *Pf*LysRS with the compound **36K3** aligned with **36** and L-lysine. Compound **36, 36K3** and L-lysine are depicted as sticks.

The binding of amino acids to their corresponding subpockets is the most distinctive feature of the aaRSs family, and the utilization of this pocket will significantly enhance the binding affinity and selectivity of lead drug molecules. For instance, mupirocin occupies the Ile binding pocket of IleRS[4], while halofuginone binds to the Pro binding pocket in ProRS[8]. Recently reported adenosine amino acid compounds also occupy amino acid binding sites through a reactive hijacking mechanism and amino acid formation conjugates[44]. Obafluorin uses o-diphenol to simulate the side hydroxyl group and main amino group of threonine to bind to a conserved zinc ion at the threonine binding site[45]. In this work, the enzyme inhibitory activity of L-lysine linked compound **36K3** was increased by 10 times and the selectivity was increased to 1250:1, indicating the value of this site in LysRS inhibitor development.

However, stronger in vitro *Pf*LysRS inhibition of compound 36K2-36K5 did not translate into better inhibition of parasite growth. There could be several reasons for this. First, previous studies have shown that the addition of an amino acid moiety has adverse effects on the membrane permeability of inhibitors[46]. The CLogP value decreased from 3.39 to 1.77 and the molecular weight increased from 429 to 572 after L-lysine was linked to compound **36.** Reducing CLogP on the one hand may help to improve solubility and bioavailability, but on the other hand may also result in a reduced ability to penetrate the cell membrane. Therefore, in future work, it will be valuable to modify the amino acid part to improve bioavailability, such as modifications in the form of amino protected prodrugs. Second, as two-site inhibitors that binds to ATP and amino acid sites, these compounds have to compete with both endogenous ATP and lysine. The binding benefit of introducing the native lysine moiety may be not sufficient to offset the disadvantage of endogenous lysine competition. The use of amino acid binding sites may require the introduction of pharmacophore that binds *Pf*LysRS at least more strongly than lysine. Third, it has been reported that there are some important

kinases in *P. falciparum* that can be used as targets for antimalarial drugs[47]. Although the receptor tyrosine kinase family to which ALK belongs does not exist in malaria, we cannot grossly rule out the possibility that well-active molecules such as **36** also inhibit the function of other *P. falciparum* kinases or even other ATP-binding proteins. In this sense, developing molecules that inhibit both *P. falciparum* aaRS and *P. falciparum* kinases may have advantages over molecules such as **36K3** that are highly selective to aaRS alone.

Both aaRSs and kinases are important drug targets, and both use ATP as a substrate. Kinase inhibitors have been intensely developed and derivatized in the last decades to potently inhibit kinase targets[48,49]. This work provides a new example of kinase inhibitor repurposing to inhibit a non-kinase target, in this case aaRS, which could help to promote the development of new aaRS inhibitors.

## Methods
### Compound synthesis
All reagents were purchased from commercial sources, and were used without further purification. Reaction products were purified by normal phase silica-gel flash column chromatography (300–400 mesh). Structure characterization was based on NMR spectroscopy, recorded on a 500 MHz Agilent system, a Bruker 400 MHz or a Bruker 300 MHz, as well as mass spectrometry, measured using maXis 4 G or Shimadzu LCMS-2010EV. Spectroscopic data was analyzed using MestReNova 11.0.1 (Mestrelab Research). The synthesis paths are shown in Supplementary Figs. 2, 9 and 10. The detailed methods for synthesis of the compounds are included in Supplementary Note 1.

### Protein expression
*Pf*LysRS (77–583) was constructed in the vector pET20b, with a C-terminal 6×His tag. The protein was expressed in BL21 (DE3) strain with 0.2 mM

**Fig. 7 | Some ASP3026 analogues improve in cellulo inhibition against the blood stage of *P. falciparum*. a** Chemical structures of compounds **34, 36, 38** and **36a**. **b** Schematic diagrams of erythrocytic-stage parasites (strain 3D7) growth inhibition experiment. The parasites were allowed to grow with the incubation of DMSO or different concentrations of ASP3026 derived compounds for 72 h. **c** The potency of compound **34** against the growth of erythrocytic-stage *P. falciparum* 3D7 parasites. **d** The potency of compound **36** against the growth of erythrocytic-stage *P. falciparum* 3D7 parasites. **e** The potency of compound **38** against the growth of erythrocytic-stage *P. falciparum* 3D7 parasites. **f** The potency of compound **36a** against the growth of erythrocytic-stage *P. falciparum* 3D7 parasites. Error bars in **c–f** represent SD of two or three technical repeats.

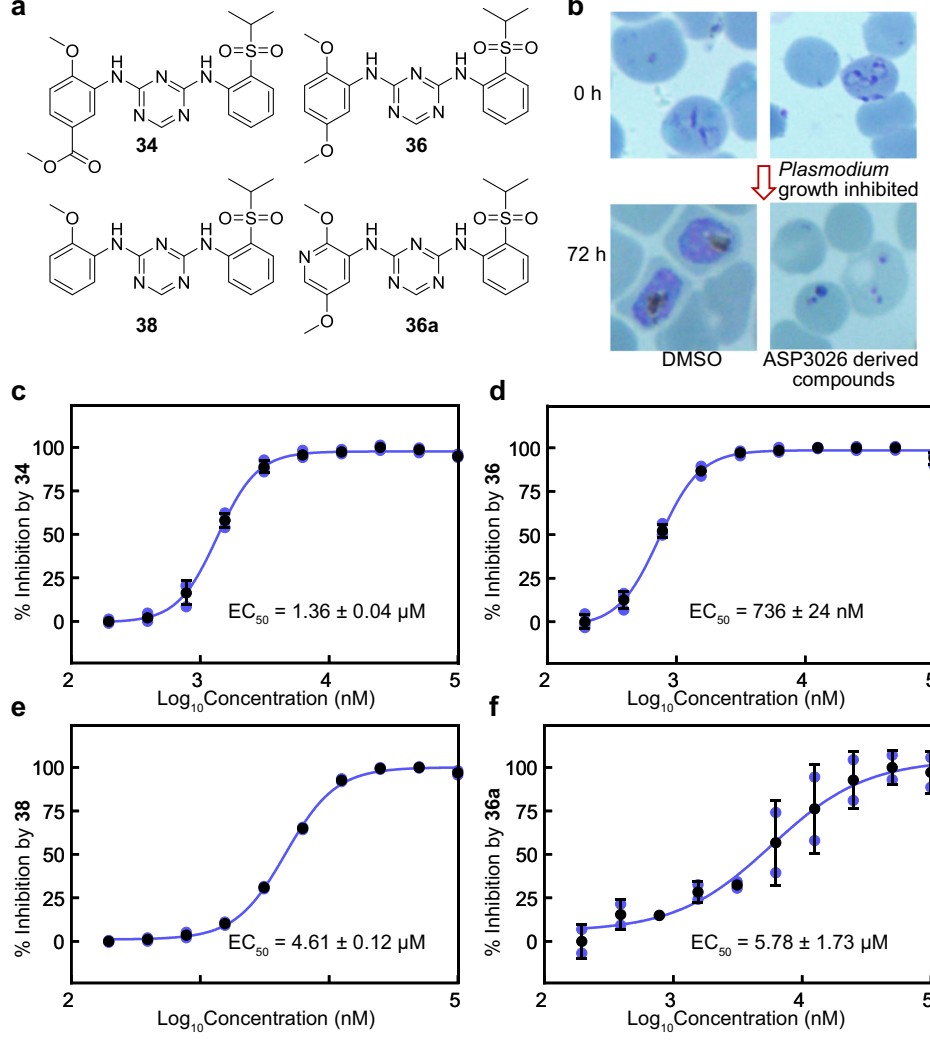

IPTG for 20 h at 16 °C. The cell pellet (from 4 to 8 L) was lysed in the wash buffer containing 500 mM NaCl, 20 mM Tris pH 8.0, and 15 mM imidazole, loaded onto a Ni-HiTrap column and washed with wash buffer. The protein was eluted with the elution buffer containing 500 mM NaCl, 20 mM Tris pH 8.0, and 250 mM imidazole. Then the protein was concentrated and further purified by gel filtration Superdex 200 column with the running buffer containing 20 mM Tris pH 8.0 and 150 mM NaCl. The peak fractions were then concentrated for activity assays or crystallization experiments. *Hs*LysRS (70–581) was constructed in the vector pET20b with a C-terminal 6×His tag, and purified similarly.

The plasmid pCDNA3.1_8×His-tev-ALK (kinase domain, 1068–1410) was constructed[50] and extracted from *E. coli* DH5α (DE3) strain by alkaline solution method. The plasmid was transiently transfected into 3 L Expi293 cells at a density of $3 \times 10^6$ cells/mL using transfection reagent PEI (25,000 Da). The cells were collected by centrifugation on the 5th day after transfection, and lysed at 600 bar at 4 °C. The centrifugation supernatant of cell lysate was loaded to a pre-balanced Ni-NTA column and washed with 100 mL solution containing 25 mM Tris pH 8.0, 150 mM NaCl, and 25 mM imidazole. The ALK protein was then eluted with 40 mL solution containing 25 mM Tris pH 8.0, 150 mM NaCl, and 250 mM imidazole. The eluent was concentrated to 500 μL and further purified by a Superdex 200 column with the running buffer containing 25 mM Hepes pH 7.4 and 250 mM NaCl. The peak fractions were used for enzymatic assays.

### Thermal shift assay

*Pf*LysRS protein was prepared at 10 μM concentration in a buffer containing 20 mM Tris pH 8.0, 200 mM NaCl, 500 μM L-lysine, and 200 μM Compound. Aliquots (18 μL) were added to a 96-well PCR plate and incubated at ambient temperature for 10 min. SYPRO Orange dye (Sigma) was diluted in the assay buffer containing 20 mM Tris pH 8.0 and 200 mM NaCl to a 40× concentration, and 2 μL of the 40× dye solution was added to the PCR plate to bring the final assay volume to 20 μL. After complete mixing, the final solutions were heated from 25 to 90 °C at a rate of 0.015 °C per sec, and fluorescence signals were monitored by QuantStudio 3 (Applied Biosystems by Thermo Fisher Scientific).

### Surface Plasmon Resonance (SPR) assay

*Pf*LysRS was chemically immobilized on a Biacore CM5 sensorchip (immobilization level ~20000 resonance unit/RU) at pH 5.0 according to the immobilization kit (GE Healthcare). *Pf*LysRS (1.7 nM) was captured for 90 s at a flow of 30 μL·min⁻¹. Affinity between *Pf*LysRS and compounds was measured by SPR on a Biacore 8 K (GE Healthcare) at 25 °C with a running buffer (0.02 M phosphate buffer pH 7.4, 2.7 mM KCl, 0.137 M NaCl, 0.05% Tween 20, with 1 mM L-lysine). The association time was 90 s, the dissociation time was 120 s, and the flowrate was 30 μL·min⁻¹. Kinetic evaluation of the interaction between LysRS and compounds was performed by global fitting of the data to a 1:1 interaction model using Biacore Evaluation Software 3.1 (GE Healthcare).

## ATP hydrolysis assays

The ATP hydrolysis assays were based on Kinase-Glo luminescent Kit (Promega). The experiment was carried out in Corning 384-well white flat bottom microplates. The buffer in the reaction contains 25 mM Hepes pH 7.4, 140 mM NaCl, 40 mM $MgCl_2$, 30 mM KCl, 1 mM TCEP, 0.1 mg·mL$^{-1}$ BSA, and 0.004% Tween-20. The compounds were diluted to different concentrations with a 5-fold gradient.

For LysRS assays, DMSO was added into negative control wells and L-lysine was removed in positive control wells. 100 nM LysRS in 2.5 µL Hepes buffer was mixed with compounds and incubated for 10 min. Then 5 µL substrate mixture (2 µM ATP, 10 µM L-lysine and 31 nM pyrophosphatase) was added to start the reaction at 37 °C and incubated for 6 h.

For ALK assays, DMSO was added into negative control wells. 100 nM ALK in 2.5 µL Hepes buffer was mixed with compounds and incubated for 10 min. Then 5 µL substrate (2 µM ATP) was added to start the reaction at 37 °C and incubated for 6 h.

When the reaction reaches the planned time, 10 µL diluted Kinase-Glo reagent (1/50 diluted with a buffer containing 50 mM Tris pH 7.5 and 5% glycerol) was added and incubated for 15 min. Luminescence was measured on the Magellan plate reader (Tecan) and nonlinear regression was performed with Prism (GraphPad).

## In vitro *Plasmodium* growth assay

Parasites were cultured in human O+ erythrocytes according to standard procedures. To prepare the >80% ring stage parasites, asynchronous cultures of parasites were pretreated with 5% sorbitol, and *P. falciparum* strain 3D7 at the mid-ring stage (6–10 h post-invasion) was used to test antimalarial effects in 96-well plates. Parasites were incubated in a 96-well plate with compound containing 1% parasitemia and, 2% hematocrit for a total volume of 200 µL. The compounds were diluted from the maximum concentration of 10 µM with a 2-fold gradient dilution. A relevant DMSO concentration was used as a negative control, and cultured erythrocytes without *Plasmodium* served as a positive control. The parasites were allowed to grow for 72 h at 37 °C with 5% $CO_2$, 5% $O_2$, and 90% $N_2$. After 72 h, 100 µL of lysis buffer (0.12 mg·mL$^{-1}$ Saponin, 0.12% Triton X-100, 30 mM Tris-HCl pH 8.0, and 7.5 mM EDTA) with 5× SYBR Green I (Invitrogen; supplied in 10000× concentration) was added to each well of the plate. The plates were then incubated for 2 h in the dark prior to reading the fluorescence signal at 485 nm excitation and 535 nm emission. The percentage of inhibition was calculated by (NC-fluorescence) × 100/(NC-PC).

## Cytotoxicity assay

HepG2 cells (National Collection of Authenticated Cell Cultures) were cultured in Minimum Essential Medium (Gibco) supplemented with 10% of fetal bovine serum, 100 U·mL$^{-1}$ penicillin, 100 U·mL$^{-1}$ streptomycin (Gibco), 2 mM L-Glutamine (Gibco), 1× NEAA (Gibco) and 1 mM Sodium Pyruvate (Gibco). The cells were hemi-depleted each week with fresh medium and maintained at $1 \sim 2 \times 10^6$ cells/mL in 6-cm dishes at 37 °C and 5% $CO_2$. Tested with MycoBlue Mycoplasma Detector D101 (Vazyme), the cells were free from mycoplasma contamination.

Cell viability was analyzed by Cell Counting Kit-8 (CCK8, ApexBio) according to the manufacturer's protocols. Cells were seeded and cultured at a density of $2.5 \times 10^3$ cells per well in 100 µL medium in 96-well microplates (Corning). Then, the cells were treated with different concentrations of compound (0, 1 µM, 3 µM, 10 µM, 100 µM, 300 µM). For **36, 38** and **36K3**, the highest concentration was 100 µM due to poor solubility. After treatment for 72 h, 10 µL of CCK-8 reagent was added to each well and then cultured for 4 h. All experiments were performed in pentaplicate. The absorbance was analyzed at 450 nm using a microplate reader (Tecan) using medium without cells as blanks. Data was processed using GraphPad Prism 9.

## Protein crystallization

All crystallizations were done by the sitting drop method. *Pf*LysRS protein (30 mg·mL$^{-1}$) was premixed with 1 mM of compound and 1 mM of L-lysine at 4 °C and was crystallized by mixing 0.4 µL of protein solution with 0.4 µL of precipitant solution, containing 0.1 M MES pH 6.0, and 15% PEG4000. After incubation at 18 °C for 3 to 7 days, crystals were flash-frozen in liquid nitrogen for data collection with the cryo solution containing 0.08 MES pH 6.0, 12% PEG4000, 20% glycerol, and 1 mM compound.

## Structure determination

The *Pf*LysRS-compound complex crystal diffraction datasets were obtained from beamlines BL02U1 and BL19U1 at Shanghai Synchrotron Radiation Facility (SSRF)[51]. All datasets were processed with Xia2[52]. The structures were solved by molecular replacement using *Pf*LysRS structure (PDB code: 7BT5) with the program Molrep[33,53]. Iterative model building and refinement were performed using Coot and Phenix[54,55]. Data collection and refinement statistics are given in Supplementary Table 1.

## Statistics and reproducibility

Thermal shift measurements were conducted in four repeats. Enzymatic assay and Plasmodium growth assay were conducted in three repeats. Cytotoxicity assay were conducted in five repeats. Acquired data are presented as the mean values ± standard deviation (SD).

## Reporting summary

Further information on research design is available in the Nature Portfolio Reporting Summary linked to this article.

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

## Acknowledgements

We gratefully acknowledge the help from the staff of beamlines BL02U1 and BL19U1 at Shanghai Synchrotron Radiation Facility. This work is supported by the National Key Research and Development Program of China grant 2022YFC2303100, National Natural Science Foundation of China grants 22277132, 22277134, 21977107, 21977108, and 21977115, Shanghai

Science and Technology Committee grant 22ZR1475000, Dr. Deli Lin (SJTU) and Dr. Jingli Hou (SJTU) for assistance in SPR testing, and the State Key Laboratory of Chemical Biology.

## Author contributions

Conceptualization: J.Z. and P.F. Methodology: J.Z., M.X., Z.H., H.Q., G.Y., Y.Q., P.L., Z.Z., J.W., W.L., and P.F. Investigation: J.Z., M.X., Z.H., H.Q., G.Y., Y.Q., P.L., X.G., L.J., J.W., W.L., and P.F. Visualization: J.Z., M.X., and P.F. Funding acquisition: L.J., J.W., W.L., and P.F. Project administration: L.J., J.W., W.L., and P.F. Supervision: L.J., J.W., W.L., and P.F. Writing – original draft: J.Z. and P.F. Writing – review & editing: J.W., W.L., and P.F.

## Competing interests

The authors declare no competing interests.
