## [Peer Review File · Communications Biology]

Reviewers' comments:

Reviewer #1 (Remarks to the Author):

The ever-growing emergence of multi-resistant strains of *P. falciparum* warrants continuous effort to both discover new druggable targets and to derivatize current drug candidates/hits in order to improve either selectivity (in a context where these drugs often bind both parasite and host proteins) or, in a second phase, to improve delivery and overall bioavailability.

In this context, Zhou, Xia, and colleagues present a body of work focusing on a previously identified ALK inhibitor, ASP3026, which they showed could be repurposed to target the PfLysRS ATP binding site in vitro and showed some efficacy in cellulo in the low μM range. The optimization approach consists in testing protein stability using thermofluor and activity through an indirect measurement of ATP consumption. In the end, in cellulo data is also shown using the erythrocytic stage of *P. falciparum*. Through complete chemical resynthesis, they generated a large panel of derivatives sharing the core sub-structure of ASP3026, minus the 4-(4-methylpiperazine) piperidine group. By removing this group and then performing slight permutations, they managed to keep the core component which competes with the ATP binding site of PfLysRS while removing elements which are essential to ALK inhibition. Through subtle modifications, they generated the compound 36 which in vitro displays high selectivity against PfLysRS compared to HsLysRS and HsALK. In cellulo, this is the most promising lead of the study as the EC50 drops into the nanomolar range.

Lastly, they also generated derivatives crosslinked to lysine mimics with the intention of bridging the lysine binding site of PfLysRS and generating a two-site inhibitor. One of these compounds (36K3) display improved in vitro potency but ultimately has a degraded in cellulo activity.

In the process, the authors co-crystallized and solved the structures of many of these new compounds together with PfLysRS, mostly in the same isomorphous crystalline system.

While this study brings new chemical compounds and structural data to the community, it also presents certain shortcomings that could be addressed.

Major comments:

- 1) The lead optimization strategy primarily relies on thermal stability measurements, frequently interpreted through affinity metrics. One should be careful with thermofluor experiments as stabilisation gains don't necessarily translate into affinity gains. As such, affinity titration experiments should be performed to re-enforce the claims made or nuance should be brought to these affinity claims.
- 2) In cellulo efficacy within erythrocytes gives no information on the toxicity of the tested compounds. It would be highly informative to have CC50 values measured on primary cells so as to have a reference of an in cellulo selectivity index for all these compounds.
- 3) It would be beneficial if the highly conserved PfLysRS and HsLysRS structural cavities were also superposed and compared. By doing so, it might be possible to elucidate how the different permutations actually do improve selectivity for compound 36.
- 4) The apparent IC50 should be re-established for ASP3026 as this is the starting point for in vitro activity optimization. The previously established IC50 was performed with another kit with a different buffer system in the nucleic acid research paper.

Minor comments:

These are general rules which could be applied for the whole document:

- 1) Tm values are sometimes given with a 2-digit point precision; I think a 1-digit precision is as precise as it should get for thermofluor.
- 2) In many cases, selectivity or inhibitory activity should be precised as "in vitro".

Such as Page 7: line 18 title 'Substitution of the 4-(4-methylpiperazinyl)piperidinyl moiety significantly

improved the selectivity against PfLysRS' should include 'in vitro.'

And Page 8 line 13: Compound 38 maintains a high species selectivity index "in vitro" when comparing human and plasmodium IC50s.

3) I would rephrase the following sentence within the introduction:

'Lysyl-tRNA synthetase (LysRS) from this family is a promising antimalarial target.' Within this family of enzymes, the Lysyl-tRNA synthetase (LysRS) constitutes a promising antimalarial target.

4) Within the introduction on aaRSs targeting, consider citing the 'Aminoacyl-tRNA synthetases as therapeutic targets' review from Kwon et al. Nat reviews drug discovery. 2019 which is somewhat more up to date than some of the other references.

5) Page 4 line 2, would rephrase

'Halofuginone (HF), a prolyl-tRNA synthetase (ProRS) inhibitor, binds to the proline and tRNA A76 sites.'

To

The febrifugine derivative Halofuginone (HF), a prolyl-tRNA synthetase (ProRS) inhibitor, binds to the proline and tRNA A76 sites.

6) Page 4 line 6, would rephrase

'HF inhibits *P. falciparum* ProRS with a dissociation constant (KD) value of 19 nM.'

to

HF inhibits *P. falciparum* proliferation in cellulo, selectively binds and inhibits recombinant PfProRS within the low nanomolar range.

7) Page 4 line 8 would replace 'in lung cancer, rectal cancer, breast cancer, etc.' by 'in several cancer models.'

8) Page 6 line 10:

You should specify that NVP-TAE684 is a type 1 inhibitor of ALK.

9) Figure 1:

pdb identifiers, although mentioned in the text, should be included in the legend. Also, it may be interesting to have a structural superposition of both ASP3026 (in PfLysRS) and NVP-TAE684 (in ALK) compounds on their own to appreciate the structural arrangements made by ASP3026 to bind PfLysRS. In panel E), 'pyrimidine' is actually a chloropyrimidine?

10) Page 10 line 20: You state that 'However, compound 38b, which also uses a pyridine ring instead of a benzene ring as a substrate, significantly decreased the activity.'

Aside from thermofluor which informs us of stabilization, you do not show any activity in this figure?

11) Page 11 line 12: Instead of saying 'but significantly enhance the compound's ability to inhibit HsLysRS' one could add that the consequence of this is a decrease in relative selectivity.

12) Page 11, line 18: Would rather rephrase binding 'activity' to 'ability.'

13) Page 12, line 3: The figure reference should be (Figure 2B, 3B and 5B) instead of '(Figures 2-5).'

14) Page 14, line 4: rephrase 'which showed more than 1250 times selectivity for ALK and 200 times selectivity for HsLysRS' to 'displaying an apparent selectivity index of 1250 for ALK and 200 for

HsLysRS.'

15) Figure 4: The grey surface background makes these figures confusing; I would suggest removing unless informative.

16) Page 14: The title 'ASP3026 analogues can potently inhibit Plasmodium parasites growth' is not really what this shows; instead, the main message is 'Some ASP3026 analogues improve in cellulose inhibition against the blood stage of *P. falciparum*.'

17) Page 15, line 17: 'the inhibitory activity of compound 36K3,' specify that the in vitro inhibitory activity...

18) Page 16, line 5: Typo as 'the' is repeated.

19) Page 16, line 16: Generally, isn't the decrease of ClogP a good thing for bioavailability, especially for in vivo testing? One thing is crossing membranes; the other is remaining soluble at functional concentrations.

20) Page 17, line 2 to 6: Wasn't the proof of concept of repurposing a kinase inhibitor to target PfLysRS already shown in your original Nucleic Acid Research publication? Isn't this rather a proof of concept of further derivatization of repurposed drugs.

21) Supplementals page 6: I wonder why the dataset of pdb 8K9X displays such a strong $I/\sigma(I)$ within the last resolution shell (value of 6.3); wasn't this dataset a bit over-truncated in terms of resolution?"

Reviewer #2 (Remarks to the Author):

It is really an interesting finding of the anti-cancer compound, ASP3026, has PfLysRS potency and developed series of Plasmodium lysyl-tRNA synthetase selective inhibitors. The work described in this paper will interest the researchers in this area and help researchers to understand better of the strategy of converting inhibition selectivity from ALK to PfLysRS. I recommend to publish this work.

The paper writing is brief and clean. But I will suggest to add the word "Supplementary" in front of ALL 'Scheme Sx', 'Table Sx' and 'Figure Sxx' which do not in the paper but from 'supplementary information' part.

The supplementary information part, there are many minor mistakes there and need to be edited/corrected:
1, page S16, 300 MHz NMR did not mention here, but NMR data from 300 MHz reported from the later part (like page S33).

2, Looks the NMR data generated by machine and many J values in the report are not reasonable

3, page S17, 6.70 (t, J = ...), j should be italic

4, page S19 line 7, compound 4 (100 mg..) should be 'compound 5'

5, page S20 line 10, 1.30 (d, J = 6.8, 6H)...should be J = 6.8 Hz...

6, page S25 top, format used as "align left", it should be evenly between the margin

7, page S25 line 6, ¹³C NMR looks have one more signal than it should be, since two C from isopropyl group at 15.4. 77.2 came from CDCl₃?

8, page S29, compound 38a NMR looks not right. The H number from high field is not right. Need double check

9, page S31, S32, S33, S34, S37, S38, S39, S41, and S41, 'To a solution of of palladium...' should be 'To a solution of palladium...'

10, page S33, S34, S35 and S37, the reaction temperature in the reaction scheme is different from the experimental description part. Should double check and correct.

- 11, page S34 36c NMR part, '7.59(t, J=8.6, 7.3, 1.7Hz)', J has 3 number?
- 12, page S37 compound 36e 1H NMR, 1.28(d, J = 2.7Hz..) is not right. Please check J=?
- 13, page S42 13C NMR part, look have too many C signal there. Should be 26. Please check
- 14, page S44 line 8, 126.54 should be 126.5
- 15, page S45 bottom compound 36K4' 13C NMR part, should be 27 C peak, but reported 28 here. Need to double check
- 16, page S46 bottom, C44H53N8O9SNa should be C44H52...?
- 17, page S47, 13C NMR part, should be 28 C peak, but reported 29. Need to check

Reviewer #3 (Remarks to the Author):

The authors previously found that the human ALK inhibitor, ASP3026, is a reasonably potent inhibitor of the plasmodium tRNA synthetase enzyme PflsRS. In this paper, using a structure guided and rational approach, they synthesize analogs with improved potency at the Pf enzyme and dramatically improved selectivity relative to human ALK. They also crystallize PflsRS with several different key compounds in the SAR to help validate their SAR conclusions. The work is likely to be of some interest to those in the field of malaria drug research, but I do have some questions/comments that I would like to see addressed, roughly in order of importance to the overall strength of the arguments presented:

- 1) Page 14-15: the authors pick up significant activity and selectivity at the enzyme by covalently linking lysine to their ASP3026 derivatives, but lose potency at parasite inhibition. They attribute this to increased polarity impeding access to the target by decreasing permeability. I would like to see some empirical evidence for that, as there are other at least equally plausible explanations for the observed drop-off in parasite inhibition. Among other explanations, it is also possible that compounds now have to compete with both endogenous ATP and lysine, both of which are at concentrations substantially higher than in their enzyme assay.
- 2) Related to point 1, for the 6 compounds I could find that have IC50 values reported for both enzyme inhibition and parasite growth inhibition in the paper (ASP3026, 34, 36, 36a, 38, and 38a), there is no discernible correlation between the 2 values. As is typically the case for ATP-competitive kinase inhibitors, ASP3026 itself hits several additional human kinases (see doi: 10.1158/1535-7163.MCT-13-0395), and it is quite possible (I would say likely) that many of the compounds in the paper's SAR inhibit several Pf kinases as well, and may in fact exert their parasite growth inhibitory effects via inhibition of multiple Pf kinases in addition to/instead of via inhibition of PflsRS. The fact that the lysine conjugates lose potency at human ALK and also lose potency in *P. falciparum* growth inhibition would be consistent with the hypothesis that the observed *P. falciparum* inhibition for compounds in the SAR is due to inhibition of multiple essential *P. falciparum* kinases. I would like to see some attempt to understand the *P. falciparum* kinase activity for at least some of the molecules in this paper, especially given the lack of correlation observed between PflsRS inhibition and *P. falciparum* growth inhibition.
- 3) Page 10-11: the difference in enzyme activity between compounds 38a and 38b is interesting. I would expect that compound 38b would be conformationally affected by the repulsion between the 2 lone pairs at the ortho positions (that is, the pyridyl and triazolyl nitrogens); have the authors looked at relative low energy conformations of the 2 molecules, and does that help to rationalize the activity difference?
- 4) Page 11: the authors make the argument that losing an H-bond interaction with Asn339 explains why compounds 36b-f lose activity at the enzyme, but it seems to me that compound 36e would still retain that H-bond. Is that right, and if so, is there some alternative explanation the authors can

propose for why this compound loses activity?

5) In the Results section, first paragraph (p. 6), the authors claim that their earlier paper ref. 32 verifies that ASP3026 is an ATP competitive inhibitor. This seems highly likely based on the crystal structure from that paper, where the compound is shown to bind in the ATP site, but I did not see any direct competition experiments with ATP; maybe I missed it? If not, it might be more appropriate to say that the compound was shown by x-ray to bind in the ATP pocket or something similar.

6) Page 8, second paragraph I think the word "parapet" should be "para"?

RE: manuscript COMMSBIO-24-0909 by Zhou *et al.*

All reviewers provided thoughtful and helpful comments on our work. We considered and discussed these comments very carefully. Listed below are point-by-point responses to each comment of each reviewer.

Reviewer 1 believed that “this study brings new chemical compounds and structural data to the community”, and raised some shortcomings that we need to address. Here are our answers to these questions.

Reviewer #1 (Remarks to the Author):

The ever-growing emergence of multi-resistant strains of *P. falciparum* warrants continuous effort to both discover new druggable targets and to derivatize current drug candidates/hits in order to improve either selectivity (in a context where these drugs often bind both parasite and host proteins) or, in a second phase, to improve delivery and overall bioavailability.

In this context, Zhou, Xia, and colleagues present a body of work focusing on a previously identified ALK inhibitor, ASP3026, which they showed could be repurposed to target the PfLysRS ATP binding site in vitro and showed some efficacy in cellulo in the low μM range. The optimization approach consists in testing protein stability using thermofluor and activity through an indirect measurement of ATP consumption. In the end, in cellulo data is also shown using the erythrocytic stage of *P. falciparum*.

Through complete chemical resynthesis, they generated a large panel of derivatives sharing the core sub-structure of ASP3026, minus the 4-(4-methylpiperazine) piperidine group. By removing this group and then performing slight permutations, they managed to keep the core component which competes with the ATP binding site of PfLysRS while removing elements which are essential to ALK inhibition.

Through subtle modifications, they generated the compound 36 which in vitro displays high selectivity against PfLysRS compared to HsLysRS and HsALK. In cellulo, this is the most promising lead of the study as the EC₅₀ drops into the nanomolar range.

Lastly, they also generated derivatives crosslinked to lysine mimics with the intention of bridging the lysine binding site of PfLysRS and generating a two-site inhibitor. One of these compounds (36K3) display improved in vitro potency but ultimately has a degraded in cellulo activity.

In the process, the authors co-crystallized and solved the structures of many of these new compounds together with PfLysRS, mostly in the same isomorphous crystalline system.

While this study brings new chemical compounds and structural data to the community, it also presents certain shortcomings that could be addressed.

Major comments:

1) The lead optimization strategy primarily relies on thermal stability measurements, frequently interpreted through affinity metrics. One should be careful with thermofluor experiments as stabilisation gains don't necessarily translate into affinity gains. As such, affinity titration experiments should be performed to re-enforce the claims made or nuance should be brought to these affinity claims.

Answer: We agree with the reviewer that the thermal shift assay is not sufficiently accurate to judge affinity. In this study, we used this assay for preliminary evaluation of compounds mainly because it is easy to operate, low cost, and can roughly estimate the binding affinity of compounds with *Pf*LysRS. In determining the choice of optimization strategy, we mainly rely on the ability of the compound to inhibit enzyme activity. As suggested by the reviewer, we added surface plasmon resonance (SPR) experiments to measure the affinity of some of the key compounds in the optimization process. The KD values between *Pf*LysRS and compounds **38**, **36**, and **36K3** are 62.8 nM, 15.9 nM, and 12.4 nM. These results were now added in *Page 8 line 10*, *Page 9 line 5*, *Page 15 line 5*, and the new *supplementary Figures S2 and S8*.

2) In *cellulo* efficacy within erythrocytes gives no information on the toxicity of the tested compounds. It would be highly informative to have CC₅₀ values measured on primary cells so as to have a reference of an *in cellulo* selectivity index for all these compounds.

Answer: We agree with the reviewer that testing for cytotoxicity is necessary for drug development. We have now tested the compounds **34**, **36**, **38**, **38a**, **36a**, and **36K3** for toxicity *in vitro* using HepG2 cells as indicators for general mammalian cell toxicity. Compound **38** exhibits an CC₅₀ of 7.7 μM, which may due to its inhibition of ALK (IC₅₀ = 532 nM). Consistent with the weak inhibition of compound **36** on ALK and *Hs*LysRS, it was weakly toxic and showed no obvious toxicity to HepG2. The cytotoxicity of **34**, **38a**, **36a**, and **36K3** was lower than that of compound **38** but higher than that of compound **36**, which may be related to their mild inhibition of ALK or *Hs*LysRS. Together with the cellular antiparasitic activity, **36** is the most promising lead compound. These results and discussion were now added in *Page 16 line 17–Page 17 line 4*, and the new *supplementary Figure S10*.

3) It would be beneficial if the highly conserved *Pf*LysRS and *Hs*LysRS structural cavities were also superposed and compared. By doing so, it might be possible to elucidate how the different permutations actually do improve selectivity for compound **36**.

Answer: As suggested by the reviewer, we have added a new *supplementary Figure S4* showing the superimposed *Pf*LysRS and *Hs*LysRS structural cavities. Compound **36** occupies a space in the ATP binding pocket of LysRS and overlap mostly with cladosporin. Most of the strongly interacting residues are conserved between *Pf*LysRS and *Hs*LysRS. It has been reported that three different residues (Thr307, Val328, and Ser344) and the unique skeleton dynamics make *Pf*LysRS more sensitive to cladosporin than *Hs*LysRS (references 26 and 42). We think the selectivity of these Asp3026 derivatives to *Pf*LysRS may be for the same reason. This discussion was added in *Page 10 line 8–14*.

References:

26 Hoepfner, D. et al. Selective and specific inhibition of the plasmodium falciparum lysyl-tRNA synthetase by the fungal secondary metabolite cladosporin. *Cell Host Microbe* 11, 654-663 (2012).

42 Chhibber-Goel, J. & Sharma, A. Side chain rotameric changes and backbone dynamics enable specific cladosporin binding in Plasmodium falciparum lysyl-tRNA synthetase. *Proteins: Structure, Function, and Bioinformatics* 87, 730-737 (2019).

4) The apparent IC₅₀ should be re-established for ASP3026 as this is the starting point for in vitro activity optimization. The previously established IC₅₀ was performed with another kit with a different buffer system in the nucleic acid research paper.

Answer: The reviewer is right. IC₅₀ is a value that is sensitive to different experimental systems. We have tested the IC₅₀ of ASP3026 in this system and added the results to the new *supplementary Figure S1*. The IC₅₀ of ASP3026 in this system is 967 ± 170 nM, which is close to /slightly higher than the previously reported IC₅₀ of 657 ± 195 nM.

Minor comments:

These are general rules which could be applied for the whole document:

1) T_m values are sometimes given with a 2-digit point precision; I think a 1-digit precision is as precise as it should get for thermofluor.

Answer: We have revised all the T_m values to 1 decimal place precision throughout the manuscript and supplementary information as suggested.

2) In many cases, selectivity or inhibitory activity should be precised as “in vitro”.

Such as Page 7: line 18 title ‘Substitution of the 4-(4-methylpiperazinyl)piperidiny moiety significantly improved the selectivity against *Pf*LysRS’ should include ‘in vitro.’

And Page 8 line 13: Compound 38 maintains a high species selectivity index “in vitro” when comparing human and plasmodium IC₅₀s.

Answer: We added "in vitro" where the reviewer pointed out and in all other similar statements in the paper.

3) I would rephrase the following sentence within the introduction:

‘Lysyl-tRNA synthetase (LysRS) from this family is a promising antimalarial target.’ Within this family of enzymes, the Lysyl-tRNA synthetase (LysRS) constitutes a promising antimalarial target.

Answer: Thanks for the suggestion. We have made this change (new Page 2 line 3).

4) Within the introduction on aaRSs targeting, consider citing the ‘Aminoacyl-tRNA synthetases as therapeutic targets’ review from Kwon et al. Nat reviews drug discovery. 2019 which is somewhat more up to date than some of the other references.

Answer: We have adopted the reviewer's suggestion and cited this review paper as reference 10 on Page 27 line 21.

5) Page 4 line 2, would rephrase

‘Halofuginone (HF), a prolyl-tRNA synthetase (ProRS) inhibitor, binds to the proline and tRNA A76 sites.’

to

The febrifugine derivative Halofuginone (HF), a prolyl-tRNA synthetase (ProRS) inhibitor, binds to the proline and tRNA A76 sites.

Answer: The sentence has been rephrased as suggested.

6) Page 4 line 6, would rephrase

‘HF inhibits *P. falciparum* ProRS with a dissociation constant (KD) value of 19 nM.’

to

HF inhibits *P. falciparum* proliferation in cellulo, selectively binds and inhibits recombinant *Pf*ProRS within the low nanomolar range.

Answer: This sentence has also been rephrased as suggested.

7) Page 4 line 8 would replace ‘in lung cancer, rectal cancer, breast cancer, etc.’ by ‘in several cancer models.’

Answer: The sentence has also been rephrased as suggested.

8) Page 6 line 10:

You should specify that NVP-TAE684 is a type 1 inhibitor of ALK.

Answer: As suggested, we have added the description of NVP-TAE684 as a type 1 inhibitor of ALK.

9) Figure 1:

pdb identifiers, although mentioned in the text, should be included in the legend. Also, it may be interesting to have a structural superposition of both ASP3026 (in *Pf*LysRS) and NVP-TAE684 (in ALK) compounds on their own to appreciate the structural arrangements made by ASP3026 to bind *Pf*LysRS. In panel E), ‘pyrimidine’ is actually a chloropyrimidine?

Answer: We have made the suggested revisions, and the superimposed ASP3026 and NVP-TAE684 is shown in new *Figure 1D*.

(D) ASP3026 in *Pf*LysRS and NVP-TAE684 in ALK are superimposed using the adenine ring of ATP as a reference.

10) Page 10 line 20: You state that ‘However, compound 38b, which also uses a pyridine ring instead of a benzene ring as a substrate, significantly decreased the activity.’

Aside from thermofluor which informs us of stabilization, you do not show any activity in this figure?

Answer: Thanks for pointing out the inaccuracy of this expression. We have revised this sentence to “significantly decreased the ability to stabilize *Pf*LysRS in the thermal shift assay” (new Page 11 line 11).

11) Page 11 line 12: Instead of saying ‘but significantly enhance the compound’s ability to inhibit *Hs*LysRS’ one could add that the consequence of this is a decrease in relative selectivity.

Answer: We have revised this sentence to “but significantly decreases the species selectivity of the

compounds” (new Page 12 line 11).

12) Page 11, line 18: Would rather rephrase binding ‘activity’ to ‘ability.’

Answer: The word has been rephrased as suggested (new Page 12 line 17).

13) Page 12, line 3: The figure reference should be (Figure 2B, 3B and 5B) instead of ‘(Figures 2-5).’

Answer: The figure reference has been changed to Figure 2B, 3B and 5B (new Page 13 line 6).

14) Page 14, line 4: rephrase ‘which showed more than 1250 times selectivity for ALK and 200 times selectivity for *HsLysRS*’ to ‘displaying an apparent selectivity index of 1250 for ALK and 200 for *HsLysRS*.’

Answer: The sentence has been rephrased as suggested (new Page 15 line 10).

15) Figure 4: The grey surface background makes these figures confusing; I would suggest removing unless informative.

Answer: The grey surface background has been removed as suggested. It is shown below for the reviewer’s convenience.

Figure 4. Cocystal structures of *PfLysRS* with the compounds 34, 35, 36 and 38.

16) Page 14: The title ‘ASP3026 analogues can potently inhibit *Plasmodium* parasites growth’ is not really what this shows; instead, the main message is ‘Some ASP3026 analogues improve in cellulo inhibition against the blood stage of *P. falciparum*.’

Answer: This title has been changed to 'Some ASP3026 analogues improve the inhibition against the blood stage of *P. falciparum*' (new Page 15 line 21).

17) Page 15, line 17: ‘the inhibitory activity of compound 36K3,’ specify that the in vitro inhibitory activity...

Answer: We have added 'in vitro' here (new Page 18 line 7) and in other places like this.

18) Page 16, line 5: Typo as ‘the’ is repeated.

Answer: The extra 'the' has been deleted.

19) Page 16, line 16: Generally, isn't the decrease of ClogP a good thing for bioavailability, especially for in vivo testing? One thing is crossing membranes; the other is remaining soluble at functional concentrations.

Answer: We agree with the reviewer that druggability depends on multiple aspects. We have revised the discussion in Page 18–19 to avoid being too judgmental. We also put this discussion below for the reviewer's convenience.

*However, stronger in vitro PfLysRS inhibition of compound 36K2-36K5 did not translate into better inhibition of parasite growth. There could be several reasons for this. First, previous studies also shown that the addition of an amino acid moiety has adverse effects on the membrane permeability of inhibitors.⁴⁷ The CLogP value decreased from 3.39 to 1.77 and the molecular weight increased from 429 to 572 after L-lysine was linked to compound 36. Reducing CLogP on the one hand may help to improve solubility and bioavailability, but on the other hand may also result in a reduced ability to penetrate the cell membrane. Therefore, in future work, it will be valuable to modify the amino acid part to improve bioavailability, such as modifications in the form of amino protected prodrugs. Second, as two-site inhibitors that binds to ATP and amino acid sites, these compounds have to compete with both endogenous ATP and lysine. The binding benefit of introducing the native lysine moiety may be not sufficient to offset the disadvantage of endogenous lysine competition. The use of amino acid binding sites may require the introduction of pharmacophore that binds PfLysRS at least more strongly than lysine. Third, it has been reported that there are some important kinases in *P. falciparum* that can be used as targets for antimalarial drugs.⁴⁸ Although the receptor tyrosine kinase family to which ALK belongs does not exist in malaria, we cannot grossly rule out the possibility that well-active molecules such as 36 also inhibit the function of other *P. falciparum* kinases or even other ATP-binding proteins. In this sense, developing molecules that inhibit both *P. falciparum* aaRS and *P. falciparum* kinases may have advantages over molecules such as 36K3 that are highly selective to aaRS alone.*

References:

47 Forrest, A. K. et al. Aminoalkyl adenylate and aminoacyl sulfamate intermediate analogues differing greatly in affinity for their cognate *Staphylococcus aureus* aminoacyl

tRNA synthetases. Bioorg Med Chem Lett 10, 1871-1874 (2000).

48 Mori, M. et al. *The selective anaplastic lymphoma receptor tyrosine kinase inhibitor ASP3026 induces tumor regression and prolongs survival in non-small cell lung cancer model mice. Molecular cancer therapeutics 13, 329-340 (2014).*

20) Page 17, line 2 to 6: Wasn't the proof of concept of repurposing a kinase inhibitor to target PflLysRS already shown in your original Nucleic Acid Research publication? Isn't this rather a proof of concept of further derivatization of repurposed drugs.

Answer: We apologize for this confusing statement. In our original Nucleic Acid Research paper, we found that ASP3026 is a dual inhibitor for aaRS and kinase. The inhibitory effect of ASP3026 on human ALK was about 200 times stronger than that of PflLysRS. What we're trying to say here is that it's practically possible to completely change its selectivity from kinase to aaRS. We have now rephrased this statement on *Page 19 line 9* as "*This work provides a new example of derivatization of kinase inhibitors repurposed to inhibit aaRS, which will help to promote the development of aaRS-targeted drugs.*".

21) Supplementals page 6: I wonder why the dataset of pdb 8K9X displays such a strong $I/\sigma(I)$ within the last resolution shell (value of 6.3); wasn't this dataset a bit over-truncated in terms of resolution?"

Answer: We appreciate that the reviewers found problems with our handling of this set of data. We reprocessed the data, and in fact a resolution of 2.35 Å was reasonable. We will contact the PDB to update the data processing results. Because it comes from the same set of data, the structural model does not change. The new statistics have been updated in the *Supplementary Table S1*.

Reviewer 2 recommended the publication of our paper and suggested a series of minor revisions. We are very grateful to the reviewer for the careful review and pointing out the problems in the writing of the paper.

Reviewer #2 (Remarks to the Author):

It is really an interesting finding of the anti-cancer compound, ASP3026, has PfLysRS pontency and developed series of Plasmodium lysyl-tRNA synthetase selective inhibotors. The work decrive in this paper will interest the researchers in this area and help researchers to understand better of the strategy of converting inhibition selectivity from ALK to PfLysRS. I recommend to publish this work.

The paper writing is brief and clean. But I will suggest to add the word "Supplementary" in front of ALL 'Scheme Sx', 'Table Sx' and 'Figure Sxx' which do not in the paper but from 'supplementary information' part.

Answer: Thanks for the suggestions. We have added the word 'Supplementary' in front of all Scheme Sx, Table Sx and Figure Sx in the manuscript.

The supplementary information part, there are many minor mistake there and need to be edit/correct:

1, page S16, 300 MHz NMR did not mention here, but NMR data from 300 MHz reported from the later part (like page S33).

Answer: We have added relevant information about the 300MHz NMR device to the manuscript and supplementary information.

2, Looks the NMR data generated by machine and many J value in the report are not reasonable

Answer: We have thoroughly reviewed the NMR data, with a particular focus on verifying the J-coupling values, and have made the necessary corrections.

3, page S17, 6.70 (t, J =...), j should be italic

Answer: The J on new page S22 has been italicized.

4, page S19 line 7, compound 4 (100 mg..) should be 'compound 5'

Answer: 'Compound 4' in new page S24 line 2 has been corrected to 'compound 5'.

5, page S20 line 10, 1.30 (d, J = 6.8, 6H)...should be J = 6.8 Hz...

Answer: The unit Hz has been added.

6, page S25 top, format used as "align left", it should be evenly between the margin

Answer: The format has been corrected.

7, page S25 line 6, ¹³C NMR looks have one more signal than it should be, since two C from isopropyl group at 15.4. 77.2 came from CDCl₃?

Answer: We are grateful to the reviewer for pointing this out. We mistakenly included carbons

from the CDCl₃ around δ 77.2, while the hydrogens at the δ 15.4 came from isopropyl group. The new NMR C spectrum has been checked and corrected.

8, page S29, compound **38a** NMR looks not right. The H number from high field is not right. Need double check

Answer: We double checked the NMR of compound **38a** and corrected it.

9, page S31, S32, S33, S34, S37, S38, S39, S41, and S41, 'To a solution of of palladium...' should be 'To a solution of palladium..'

Answer: All these typos have been corrected as suggested.

10, page S33, S34, S35 and S37, the reaction temperature in the reaction scheme is different from the experimental description part. Should double check and correct.

Answer: We checked the original records and the reaction temperature should be 105 °C. The temperature in the corresponding reaction scheme has been modified.

11, page S34 36c NMR part, '7.59(t, J=8.6, 7.3, 1.7Hz)', J has 3 number?

Answer: After examination, we found that the peak split was complicated, and it was changed to m.

12, page S37 compound 36e 1H NMR, 1.28(d, J = 2.7Hz..) is not right. Please check J=?

Answer: We checked and corrected the J value to 6.8 Hz.

13, page S42 13C NMR part, look have too many C signal there. Should be 26. Please check

Answer: We double checked compound 36K3 13C NMR and corrected it.

14, page S44 line 8, 126.54 should be 126.5

Answer: The number of decimal places for this value has been corrected.

15, page S45 bottom compound 36K4' 13C NMR part, should be 27 C peak, but reported 28 here. Need to double check

Answer: We double checked compound 36K4' 13C NMR and corrected it.

16, page S46 bottom, C₄₄H₅₃N₈O₉SNa should be C₄₄H₅₂...?

Answer: Thanks for the suggestions. It has been corrected to C₄₄H₅₂N₈O₉SNa⁺.

17, page S47, 13C NMR part, should be 28 C peak, but reported 29. Need to check

Answer: After inspection, the peak at δ 117.4 was pointed by mistake. It has now been deleted.

Reviewer 3 found our paper interesting in the field of antimalarial drug development, and made some very insightful comments. We have carefully revised the paper based on these comments, especially the discussion section.

Reviewer #3 (Remarks to the Author):

The authors previously found that the human ALK inhibitor, ASP3026, is a reasonably potent inhibitor of the plasmodium tRNA synthetase enzyme PfLysRS. In this paper, using a structure guided and rational approach, they synthesize analogs with improved potency at the Pf enzyme and dramatically improved selectivity relative to human ALK. They also crystallize PfLysRS with several different key compounds in the SAR to help validate their SAR conclusions. The work is likely to be of some interest to those in the field of malaria drug research, but I do have some questions/comments that I would like to see addressed, roughly in order of importance to the overall strength of the arguments presented:

1) Page 14-15: the authors pick up significant activity and selectivity at the enzyme by covalently linking lysine to their ASP3026 derivatives, but lose potency at parasite inhibition. They attribute this to increased polarity impeding access to the target by decreasing permeability. I would like to see some empirical evidence for that, as there are other at least equally plausible explanations for the observed drop-off in parasite inhibition. Among other explanations, it is also possible that compounds now have to compete with both endogenous ATP and lysine, both of which are at concentrations substantially higher than in their enzyme assay.

Answer: This comment is very pertinent and enlightening. The incomplete correlation between the enzyme inhibitory activity and antimalarial activity of the compounds is discussed more comprehensively as suggested by the reviewers on *Page 18–19*. We also include the revised discussion below for the reviewer's convenience.

*However, stronger in vitro PfLysRS inhibition of compound 36K2-36K5 did not translate into better inhibition of parasite growth. There could be several reasons for this. First, previous studies have shown that the addition of an amino acid moiety has adverse effects on the membrane permeability of inhibitors.⁴⁷ The CLogP value decreased from 3.39 to 1.77 and the molecular weight increased from 429 to 572 after L-lysine was linked to compound 36. Reducing CLogP on the one hand may help to improve solubility and bioavailability, but on the other hand may also result in a reduced ability to penetrate the cell membrane. Therefore, in future work, it will be valuable to modify the amino acid part to improve bioavailability, such as modifications in the form of amino protected prodrugs. Second, as two-site inhibitors that binds to ATP and amino acid sites, these compounds have to compete with both endogenous ATP and lysine. The binding benefit of introducing the native lysine moiety may be not sufficient to offset the disadvantage of endogenous lysine competition. The use of amino acid binding sites may require the introduction of pharmacophore that binds PfLysRS at least more strongly than lysine. Third, it has been reported that there are some important kinases in *P. falciparum* that can be used as targets for antimalarial drugs.⁴⁸ Although the receptor tyrosine kinase family to which ALK belongs does not exist in malaria, we cannot grossly rule out the possibility that well-active molecules such as 36 also inhibit the function*

of other *P. falciparum* kinases or even other ATP-binding proteins. In this sense, developing molecules that inhibit both *P. falciparum* aaRS and *P. falciparum* kinases may have advantages over molecules such as **36K3** that are highly selective to aaRS alone.

References:

47 Forrest, A. K. et al. Aminoalkyl adenylate and aminoacyl sulfamate intermediate analogues differing greatly in affinity for their cognate *Staphylococcus aureus* aminoacyl tRNA synthetases. *Bioorg Med Chem Lett* 10, 1871-1874 (2000). [https://doi.org/10.1016/S0960-894X\(00\)00360-7](https://doi.org/10.1016/S0960-894X(00)00360-7)

48 Mori, M. et al. The selective anaplastic lymphoma receptor tyrosine kinase inhibitor ASP3026 induces tumor regression and prolongs survival in non-small cell lung cancer model mice. *Molecular cancer therapeutics* 13, 329-340 (2014). <https://doi.org/10.1158/1535-7163.mct-13-0395>.

2) Related to point 1, for the 6 compounds I could find that have IC50 values reported for both enzyme inhibition and parasite growth inhibition in the paper (ASP3026, 34, 36, 36a, 38, and 38a), there is no discernible correlation between the 2 values. As is typically the case for ATP-competitive kinase inhibitors, ASP3026 itself hits several additional human kinases (see doi: 10.1158/1535-7163.MCT-13-0395), and it is quite possible (I would say likely) that many of the compounds in the paper's SAR inhibit several *Pf* kinases as well, and may in fact exert their parasite growth inhibitory effects via inhibition of multiple *Pf* kinases in addition to/independent of via inhibition of *Pf*LysRS. The fact that the lysine conjugates lose potency at human ALK and also lose potency in *P. falciparum* growth inhibition would be consistent with the hypothesis that the observed *P. falciparum* inhibition for compounds in the SAR is due to inhibition of multiple essential *P. falciparum* kinases. I would like to see some attempt to understand the *P. falciparum* kinase activity for at least some of the molecules in this paper, especially given the lack of correlation observed between *Pf*LysRS inhibition and *P. falciparum* growth inhibition.

Answer: This comment, like the previous one, is very instructive. We really cannot rule out the possibility that some of these compounds may inhibit certain kinases in the malaria parasite. The revised discussion is included in the reply to the comment 1 above.

3) Page 10-11: the difference in enzyme activity between compounds 38a and 38b is interesting. I would expect that compound 38b would be conformationally affected by the repulsion between the 2 lone pairs at the ortho positions (that is, the pyridyl and triazolyl nitrogens); have the authors looked at relative low energy conformations of the 2 molecules, and does that help to rationalize the activity difference?

Answer: The reviewer again made a great point. We calculated the local minimum energy conformations of compounds **38a** and **38b** based on the binding conformation of compound **38** in crystal structure and it did help to rationalize the activity difference. The DFT calculations results were added to new *Supplementary Figure S5*. In compound **38a**, there is a repulsion between one CH in the pyridine ring and the other CH in the benzene ring, resulting in the three aromatic rings of the compound tending not to be in the same plane. This is the same as compound **38**, so the minimum energy conformation of compound **38a** is almost the same as the *Pf*LysRS protein binding conformation. But in compound **38b**, CH at this position in the pyridine ring changes to N.

In the minimum energy conformation of **38b**, the three aromatic rings converge to the same plane due to conjugation. There is a significant difference between its minimum energy conformation and the protein binding conformation ($\Delta G^{\text{pro}} - \Delta G^{\text{min}} = 6.1$ kcal/mol). Therefore, the minimum energy conformation of **38b** is not conducive to binding *Pf*LysRS. This might be reason for the activity difference between the compounds **38a** and **38b**. These discussions have been added to Page 11 line 11–22 in the manuscript.

Figure S5. Effect of aromatic ring substitutions on the conformation of compounds.

4) Page 11: the authors make the argument that losing an H-bond interaction with Asn339 explains why compounds 36b-f lose activity at the enzyme, but it seems to me that compound 36e would still retain that H-bond. Is that right, and if so, is there some alternative explanation the authors can propose for why this compound loses activity?

Answer: We agree with the reviewer that some compounds may retain hydrogen-bond interactions with proteins to some extent. In addition to hydrogen bond interactions, triazine also had stacking interactions with the class II aaRS conserved Phe342 residue. All the compounds modified with triazine showed a significant decrease in binding strength. Inspired by the difference between compounds **38a** and **38b**, when **36e** changes the N atom in triazine to CH, it also increases the repulsion with CH in the other two benzene rings. Therefore, the minimum energy conformation of **36e** is not conducive to binding with the protein as shown in *Supplementary Figure S5C*. Related discussion has been changed to 'these changes may weaken or even lose of hydrogen bonding interactions with Asn339 residues and stacking interactions with Phe342 residues. Some changes such as changes in **36e** will also increase the repulsion between aromatic rings, affect the conformation between the three aromatic rings, resulting in the reduced binding.' (Page 12 line 19–22).

5) In the Results section, first paragraph (p. 6), the authors claim that their earlier paper ref. 32 verifies that ASP3026 is an ATP competitive inhibitor. This seems highly likely based on the crystal structure from that paper, where the compound is shown to bind in the ATP site, but I did not see any direct competition experiments with ATP; maybe I missed it? If not, it might be more appropriate to say that the compound was shown by x-ray to bind in the ATP pocket or something similar.

Answer: We tried to add ATP or ATP competing inhibitor cladosporin to the buffer of SPR experiment, and found that the resonance unit (RU) of ASP3026 binding *PfLysRS* was significantly reduced. Therefore, we proposed that ASP3026 is an ATP competitive inhibitor of *PfLysRS* in our earlier paper.

6) Page 8, second paragraph I think the word “parapet” should be “para”?

Answer: The word 'parapet' has been changed to 'para position'.

REVIEWERS' COMMENTS:

Reviewer #1 (Remarks to the Author):

In response to the initial concerns raised in the first round of revisions, Zhou, Xia, and colleagues have addressed every point raised and made considerable efforts to answer experimentally or correct all suggestions. I have no further concerns in the results sections.

Minor suggestion in the newly proposed discussion:

Page 18, Line 22,

« Third, it has been reported that there are some important kinases in *P. falciparum* that can be used as targets for antimalarial drugs.⁴⁸ » Reference 48 does not seem pertinent for that statement. You could instead use the Arendse et al review on the subject (DOI: 10.1021/acscinfecdis.0c00724)

Page 19, line 8 « The kinase inhibitor drugs have been a great success in recent years. This work provides a new example of derivatization of kinase inhibitors repurposed to inhibit aaRS, which will help to promote the development of aaRS-targeted drugs. »

You could rather state: Kinase inhibitors have been intensely developed and derivatized in in the last decades to potently inhibit kinase targets. This work provides a new example of kinase inhibitor repurposing to inhibit a non-kinase target, in this case aaRS, which could help to promote the development of new aaRS inhibitors.

Reviewer #2 (Remarks to the Author):

I saw all questions from reviewers has been properly answered and stated. No further questions.

Reviewer #3 (Remarks to the Author):

This is a review of the revised manuscript. I greatly appreciate the authors' very clear rebuttal letter and the diligence they showed in the modifications/changes they made to the manuscript based on my questions/concerns. The revised manuscript addresses all my concerns and is acceptable for publication.

RE: manuscript COMMSBIO-24-0909A by Zhou *et al.*

All three reviewers have approved our revised manuscript. We would like to thank the reviewers again for their constructive comments in the previous round of review.

Reviewer 1 put forward two minor suggestions, which we have adopted and made corresponding modifications in the paper.

Reviewer #1 (Remarks to the Author):

In response to the initial concerns raised in the first round of revisions, Zhou, Xia, and colleagues have addressed every point raised and made considerable efforts to answer experimentally or correct all suggestions. I have no further concerns in the results sections.

Minor suggestion in the newly proposed discussion:

Page 18, Line 22,

« Third, it has been reported that there are some important kinases in *P. falciparum* that can be used as targets for antimalarial drugs.⁴⁸ » Reference 48 does not seem pertinent for that statement.

You could instead use the Arendse *et al* review on the subject (DOI: 10.1021/acsinfecdis.0c00724)

Response: We have replaced the previous inappropriate reference with the reference suggested by the reviewer.

Page 19, line 8 « The kinase inhibitor drugs have been a great success in recent years. This work provides a new example of derivatization of kinase inhibitors repurposed to inhibit aaRS, which will help to promote the development of aaRS-targeted drugs. »

You could rather state: Kinase inhibitors have been intensely developed and derivatized in the last decades to potently inhibit kinase targets. This work provides a new example of kinase inhibitor repurposing to inhibit a non-kinase target, in this case aaRS, which could help to promote the development of new aaRS inhibitors.

Response: We have revised this statement as suggested by the reviewer.